# A concerted mechanism involving ACAT and SREBPs by which oxysterols deplete accessible cholesterol to restrict microbial infection

David B Heisler[1†], Kristen A Johnson[2†‡], Duo H Ma[1,2], Maikke B Ohlson[1], Lishu Zhang[1], Michelle Tran[2], Chase D Corley[2], Michael E Abrams[1], Jeffrey G McDonald[2], John W Schoggins[1], Neal M Alto[1*], Arun Radhakrishnan[2*]

[1]Department of Microbiology, The University of Texas Southwestern Medical Center, Dallas, United States; [2]Department of Molecular Genetics, The University of Texas Southwestern Medical Center, Dallas, United States

*For correspondence:
neal.alto@utsouthwestern.edu (NMA);
arun.radhakrishnan@utsouthwestern.edu (AR)

[†]These authors contributed equally to this work

Present address: [‡]Fred Hutchinson Cancer Center, Clinical Research Division, Seattle, United States

**Abstract** Most of the cholesterol in the plasma membranes (PMs) of animal cells is sequestered through interactions with phospholipids and transmembrane domains of proteins. However, as cholesterol concentration rises above the PM's sequestration capacity, a new pool of cholesterol, called accessible cholesterol, emerges. The transport of accessible cholesterol between the PM and the endoplasmic reticulum (ER) is critical to maintain cholesterol homeostasis. This pathway has also been implicated in the suppression of both bacterial and viral pathogens by immunomodulatory oxysterols. Here, we describe a mechanism of depletion of accessible cholesterol from PMs by the oxysterol 25-hydroxycholesterol (25HC). We show that 25HC-mediated activation of acyl coenzyme A: cholesterol acyltransferase (ACAT) in the ER creates an imbalance in the equilibrium distribution of accessible cholesterol between the ER and PM. This imbalance triggers the rapid internalization of accessible cholesterol from the PM, and this depletion is sustained for long periods of time through 25HC-mediated suppression of SREBPs and continued activation of ACAT. In support of a physiological role for this mechanism, 25HC failed to suppress Zika virus and human coronavirus infection in ACAT-deficient cells, and *Listeria monocytogenes* infection in ACAT-deficient cells and mice. We propose that selective depletion of accessible PM cholesterol triggered by ACAT activation and sustained through SREBP suppression underpins the immunological activities of 25HC and a functionally related class of oxysterols.

## Editor's evaluation

This paper provides important and fundamental insights into the mechanisms by which 25-hydroxycholesterol, which is known to be rapidly produced in macrophages and other cells during acute infections, acts to protect cells and animals from infectious processes. The authors provide compelling evidence that the cholesterol esterifying enzyme acylCoA:cholesterol acyltransferase (ACAT) that is induced by 25-hydroxycholesterol promotes depletion of an accessible pool of plasma membrane cholesterol, producing anti-microbial effects. The work will be of interest to those studying infection and cholesterol regulation of cellular processes.

## Introduction

Cholesterol is an essential component of the membranes of animal cells and its levels are tightly regulated by multiple feedback mechanisms that control its production, uptake, and intracellular distribution (*Brown et al., 2018*; *Luo et al., 2020*). Dysregulation of cellular cholesterol homeostasis is implicated in many human diseases ranging from atherosclerosis to cancer (*Goldstein and Brown, 2015*; *Shimano and Sato, 2017*). There is also growing evidence that membrane cholesterol modulation plays key roles in host defense against bacterial and viral pathogens (*Cyster et al., 2014*; *Abrams et al., 2020a*; *Zhou et al., 2020*; *Wang et al., 2020*). While the proteins and pathways that mediate cholesterol regulation have been extensively studied, how these processes are rapidly altered during infection remains poorly defined.

Most of a mammalian cell's cholesterol is concentrated in the plasma membrane (PM), where the molecule makes up 40–50% of the total lipids (*van Meer et al., 2008*). The steady state levels of PM cholesterol are regulated by balancing the flow of cholesterol into and out of the PM. Inflows to the PM originate from two locations – (i) the endoplasmic reticulum (ER) where cholesterol is synthesized; and (ii) lysosomes where cholesterol is liberated from low-density lipoproteins (LDL) that bind to LDL receptors on the PM and are internalized by receptor-mediated endocytosis (*Brown and Goldstein, 1986*). Upon arrival at the PM from these locations, the incoming cholesterol forms complexes with sphingomyelin (SM) and other PM phospholipids, which imparts the PM with structural properties required for proper cellular function and growth (*Simons and Ikonen, 2000*; *McConnell and Radhakrishnan, 2003*). Cholesterol in excess of the sequestering capacity of PM phospholipids forms a pool that has been termed 'accessible cholesterol' (*Das et al., 2014*). Expansion of this pool signals cholesterol overaccumulation and triggers flow of accessible cholesterol out of the cell by PM cholesterol efflux proteins (*Venkateswaran et al., 2000*; *Repa and Mangelsdorf, 2000*), or into the cell by cholesterol transporters that move the lipid to organelles such as the ER (*Lange et al., 2004*; *Infante and Radhakrishnan, 2017*).

The accessible cholesterol that is transported to the ER interacts with two regulatory membrane proteins, a cholesterol sensor called Scap (*Radhakrishnan et al., 2004*; *Sokolov and Radhakrishnan, 2010*) and a cholesterol-modifying enzyme called acyl coenzyme A: cholesterol acyltransferase (ACAT), also designated as sterol O-acyltransferase (SOAT) (*Chang et al., 2006*; *Xu and Tabas, 1991*). Binding of cholesterol to Scap prevents the activation of sterol regulatory element-binding proteins (SREBPs), transcription factors that upregulate genes for lipid production, including those for cholesterol synthesis and uptake (*Brown et al., 2018*). The ACAT enzyme converts some of the incoming cholesterol to cholesteryl esters for storage in cytoplasmic lipid droplets (*Chang et al., 1997*). Thus, the transcriptional and enzymatic responses induced by accessible cholesterol arriving at the ER combine to reduce cellular cholesterol and protect the PM from cholesterol overaccumulation.

While accessible cholesterol in the PM plays a crucial role in maintaining cholesterol homeostasis and regulating cell growth, this pool of cholesterol can also be a vulnerability as it is exploited by numerous bacteria and viruses to promote infection (*Abrams et al., 2020a*; *Zhou et al., 2020*; *Wang et al., 2020*). Fortunately, animal cells have devised a clever solution to rectify this vulnerability. The solution involves an innate immune response that stimulates the expression of cholesterol 25-hydroxylase (CH25H), an enzyme that attaches a hydroxyl group to the iso-octyl sidechain of cholesterol to generate 25-hydroxycholesterol (25HC), a potent signaling molecule (*Lund et al., 1998*). 25HC produced by macrophage cells in response to infection acts in a paracrine manner to induce rapid depletion of accessible cholesterol, but not cholesterol in complexes with SM, from the PMs of nearby cells. Such depletion prevents the spread of infection by bacterial pathogens such as *Listeria monocytogenes* and *Shigella flexneri* (*Abrams et al., 2020a*) as well as cellular infection by viruses such as SARS-CoV-2, the causative agent of COVID-19 (*Wang et al., 2020*). In addition, 25HC acts in an autocrine manner to prevent macrophage lysis by bacterial pore-forming toxins that target accessible cholesterol in membranes (*Zhou et al., 2020*).

Establishing the mechanisms of how 25HC rapidly depletes accessible cholesterol from PMs has been complicated due to this oxysterol's actions on multiple cholesterol homeostatic pathways, including (*i*) binding to Insigs, which suppresses the activation of SREBPs and reduces cholesterol synthesis and uptake (*Radhakrishnan et al., 2007*); (*ii*) activating Liver X receptors (LXRs), transcription factors that upregulate genes encoding proteins involved in cholesterol efflux (*Venkateswaran et al., 2000*; *Janowski et al., 1996*; *Costet et al., 2000*); and (*iii*) stimulating the activity of ACAT,

which esterifies cholesterol (*Chang et al., 1997*; *Brown et al., 1975*). Each of these actions would lower levels of total cellular cholesterol, which would then be expected to lower accessible cholesterol in PMs. The recent studies on 25HC's role in restricting pathogen infection (*Abrams et al., 2020a*; *Zhou et al., 2020*; *Wang et al., 2020*) have pointed to the involvement of two of these pathways – ACAT and SREBP – in depleting accessible cholesterol from PMs; however, several questions remain unanswered. First, what is the primary target of 25HC that initiates depletion of accessible cholesterol from the PM? Second, what is the underlying mechanism by which 25HC depletes PMs of accessible cholesterol? Third, how does 25HC sustain low levels of accessible PM cholesterol over long time periods necessary to disrupt the lifecycles of pathogens in vivo? Finally, does 25HC inhibit both bacterial and viral infections through a common mechanism?

Here, using a panel of oxysterols with diverse structures and cell lines deficient in key components that regulate cholesterol homeostasis, we find that rapid depletion of accessible cholesterol from PMs by 25HC is solely determined by ACAT activity. Stimulation of ACAT by 25HC siphons off cholesterol that arrives at the ER for conversion to cholesteryl esters, forcing a redistribution of cellular cholesterol, which ultimately leads to a decrease in accessible PM cholesterol. Once initiated through ACAT, the depletion of accessible PM cholesterol persists for longer periods through 25HC's inhibition of transcriptional pathways responsible for cholesterol synthesis and uptake as well as 25HC's continued activation of ACAT. In the absence of ACAT, 25HC no longer protects cells from lysis by cytolysins secreted by *Clostridium perfringens* and *Bacillus anthracis*, infection by *Listeria monocytogenes*, or infection by Zika virus and coronaviruses. Moreover, in an ACAT-deficient animal model, 25HC-mediated protection of the spleen and liver from infection by *Listeria monocytogenes* is reduced. Together, these studies unite the disparate reports of antibacterial and antiviral properties of 25HC through a common mechanism orchestrated by ACAT.

## Results

### Side-chain oxysterols like 25HC trigger rapid depletion of accessible cholesterol from PMs

To carry out a detailed examination of the effects of 25HC on PM cholesterol, we chose Chinese hamster ovary (CHO) cells as a model cellular system since these cells have been used for more than two decades to study cholesterol regulatory pathways (*Chang et al., 1997*; *Goldstein et al., 2002*). To detect accessible cholesterol in PMs of live cells, we used domain 4 of anthrolysin O (ALOD4), a non-lytic protein sensor that specifically detects this pool of cholesterol in membranes (*Figure 1—figure supplement 1A*; *Infante and Radhakrishnan, 2017*; *Gay et al., 2015*). When a line of CHO cells designated CHO-K1 were grown in cholesterol-rich fetal calf serum (FCS), their PMs contained high levels of accessible cholesterol that were readily detected by ALOD4 (*Figure 1A and B*, *lane 1 of top panels*). When these cells were incubated with 25HC for increasing times, ALOD4 binding declined sharply after 1 hr and was eliminated after 4 hr (*Figure 1A*, *top panel*). When treated with increasing concentrations of 25HC, ALOD4 binding was sharply reduced after treatment with 1 µM 25HC and eliminated when 25HC exceeded 2.5 µM (*Figure 1B*, *top panel*). Thus, treatment of CHO-K1 cells with 25HC rapidly depleted accessible cholesterol from PMs in a dose-dependent manner, as observed previously in other cell types (*Abrams et al., 2020a*; *Zhou et al., 2020*).

We next asked whether 25HC was unique among oxysterols in triggering such rapid depletion of accessible PM cholesterol. To address this question, we assayed a panel of structurally diverse oxysterols (*Figure 1C*) harboring hydroxyl groups on the steroid nucleus (*A – C*) or the iso-octyl side-chain (*D – J*), as well as an oxysterol where the 3β-hydroxyl was replaced with a sulfate group (*K*). As shown in *Figure 1D* and *Figure 1—figure supplement 1B*, ALOD4 binding to the PMs of CHO-K1 cells was markedly reduced by treatment for 4 hr with 25HC (*I*), as well as oxysterols harboring hydroxyl groups at carbons 20 (*D*) or 27 (*J*). Oxysterols with hydroxyl groups at carbon 24 (*G*, *H*) partially reduced ALOD4 binding, whereas oxysterols with hydroxyl groups on carbon 22 (*E*, *F*) had no effect. These data indicate that the location of the hydroxyl modification on the sterol side-chain plays a crucial role in controlling accessible PM cholesterol. Moreover, the ability of a side-chain oxysterol like 25HC (*I*) to reduce ALOD4 binding required the 3β-hydroxyl group since replacement of this group with a sulfate group (*K*) abolished this effect. No reduction of ALOD4 binding was detected with oxysterols bearing hydroxyl groups on the steroid nucleus at carbons 4 (*A*), 7 (*B*), or 19 (*C*). While oxysterols *D*,

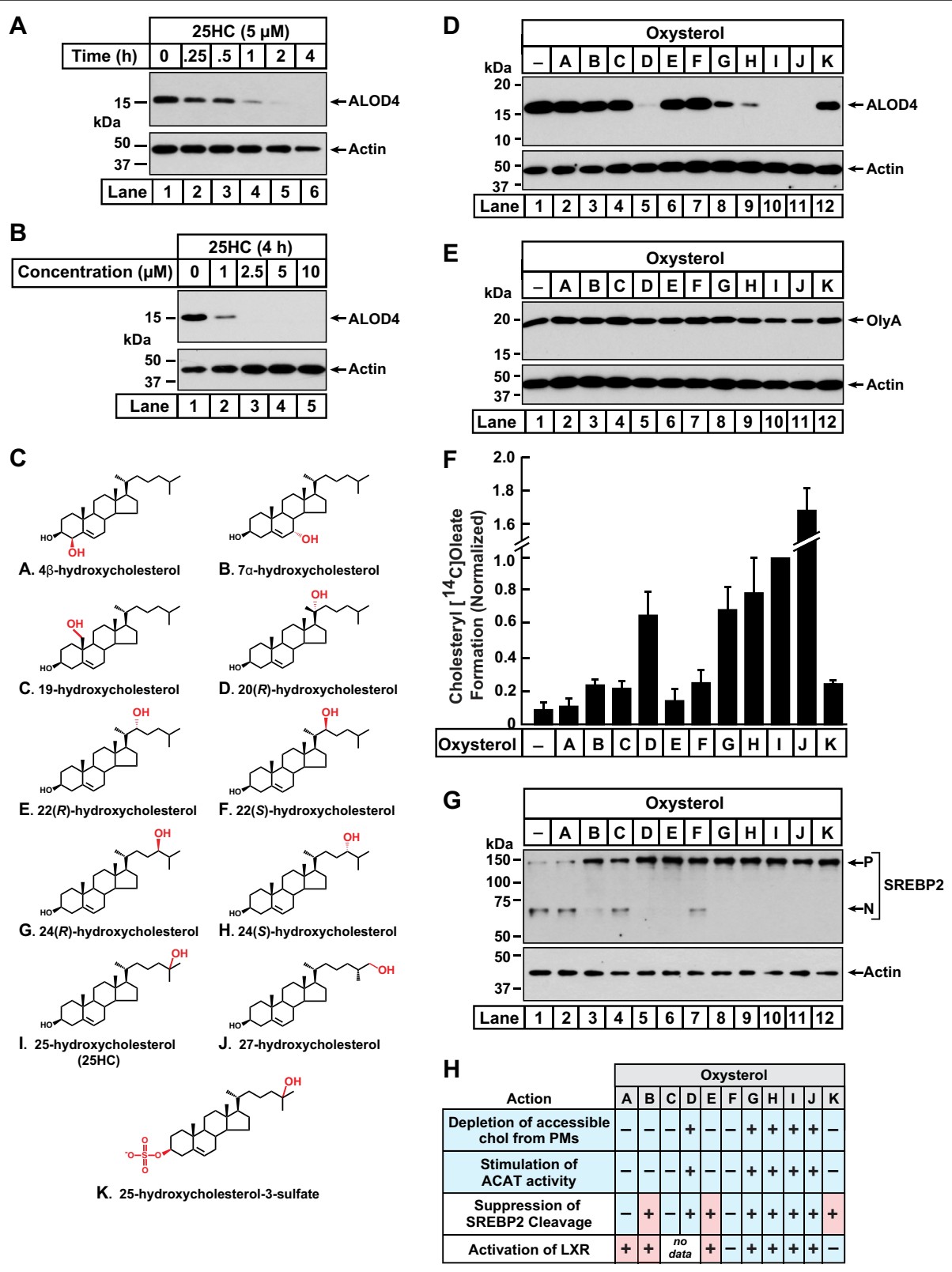

**Figure 1.** Comparison of oxysterol specificities for effects on PM cholesterol pools, stimulation of ACAT activity, and suppression of SREBP-2 cleavage. (**A, B**) Time course and dose curve analysis of 25HC treatment. On day 0, CHO-K1 cells were set up in medium B at a density of 6x10⁴ cells per well of a 48-well plate. On day 1, media was removed, cells were washed twice with 500 μl of PBS followed by addition of 200 μl of medium B supplemented with either 5 μM of 25HC (**A**) or the indicated concentrations of 25HC (**B**). After incubation at 37 °C for either the indicated times (**A**) or 4 h (**B**), media

*Figure 1 continued on next page*

*Figure 1 continued*

was removed and replaced with 200 µl of medium B supplemented with 3 µM His$_6$-Flag-ALOD4. After incubation at 37 °C for 30 min, cells were washed twice with 500 µl of PBS, harvested, and equal aliquots of cell lysates were subjected to immunoblot analysis as described in *Materials and Methods*. (**C**) Chemical structures of oxysterols tested in this study. Differences from cholesterol are highlighted in *red*. (**D, E**) Effects on PM cholesterol pools. On day 0, CHO-K1 cells were set up in medium B at a density of 6x10$^4$ cells per well of a 48-well plate. On day 1, media was removed, cells were washed twice with 500 µl of PBS followed by addition of 200 µl of medium B supplemented with 5 µM of the indicated oxysterol. After incubation at 37 °C for 4 hr, media was removed and replaced with 200 µl of medium B supplemented with 3 µM of either His$_6$-Flag-ALOD4 (**D**, *top panel*) or OlyA-His$_6$ (**E**, *top panel*). After incubation at 37 °C for 30 min, cells were washed twice with 500 µl of PBS, harvested, and equal aliquots of cell lysates were subjected to immunoblot analysis as described in *Materials and methods*. Quantification of the immunoblots are shown in *Figure 1—figure supplement 1*. (**F**) ACAT activity. On day 0, CHO-K1 cells were set up in medium B at a density of 2.5x10$^5$ cells per 60 mm dish. On day 2, media was removed, cells were washed twice with 1 ml of PBS followed by addition of 2 ml of cholesterol-depleting medium C. On day 3, media was removed, cells were washed with 1 ml of PBS followed by addition of 1 ml of medium C supplemented with 5 µM of the indicated oxysterol. After incubation at 37 °C for 1 hr, each dish was supplemented with 0.2 mM sodium [$^{14}$C]oleate (6500 dpm/nmol) and incubated at 37 °C for an additional 2 hr, after which cells were harvested, and levels of cholesteryl [$^{14}$C]oleate were measured as described in *Materials and methods*. Each column represents the mean of cholesterol esterification measurements from three or more independent experiments, and error bars show the standard error. The mean value for cholesterol esterification obtained after 25HC treatment (3.18 nmol/mg/h; n=6; standard error = ± 0.49 nmol/mg/h) was set to 1 and all other values were normalized relative to this set-point. (**G**) SREBP-2 cleavage. On day 0, CHO-K1 cells were set up in medium B at a density of 6x10$^4$ cells per well of a 48-well plate. On day 1, media was removed, cells were washed twice with 500 µl of PBS followed by addition of 200 µl of cholesterol-depleting medium C supplemented with 1% (w/v) HPCD. After incubation at 37 °C for 1 hr, media was removed, cells were washed twice with 500 µl of PBS and then treated with 200 µl of medium C supplemented with 5 µM of the indicated oxysterol. After incubation at 37 °C for 4 hr, cells were washed twice with 500 µl of PBS, harvested, and equal aliquots of cell lysates were subjected to immunoblot analysis as described in *Materials and Methods. P*, precursor form of SREBP2; *N*, cleaved nuclear form of SREBP2. Quantification of the immunoblots are shown in *Figure 1—figure supplement 5*. (**H**) Summary of oxysterol specificities for depletion of accessible cholesterol from PMs, suppression of SREBP2 cleavage, activation of LXR transcription factors, and stimulation of ACAT activity. The degree of effect (maximal or minimal) is denoted by +and –, respectively. Specificities that are different from that for depletion of accessible cholesterol from PMs (first row) are shaded *red*.

The online version of this article includes the following source data and figure supplement(s) for figure 1:

**Source data 1.** Western blots corresponding to *Figure 1A, B, D, E and G*.

**Source data 2.** Data supporting *Figure 1F* and *Figure 1—figure supplements 1B, C*; *3A, B*; *4B*; and *5*.

**Figure supplement 1.** Comparison of effects of oxysterols on PM cholesterol pools – immunoblot analysis.

**Figure supplement 2.** Comparison of effects of oxysterols on PM cholesterol pools – fluorescence microscopy analysis.

**Figure supplement 3.** Comparison of effects of oxysterols on accessible PM cholesterol levels – immunoblot analysis.

**Figure supplement 4.** Treatment of red blood cells with oxysterols does not affect levels of accessible cholesterol in their membranes.

**Figure supplement 5.** Quantification of immunoblot analysis shown in *Figure 1G*.

G, H, I, and J reduced ALOD4 binding in an all-or-none fashion, they had no detectable effect on the binding of Ostreolysin A (OlyA) (*Figure 1E* and *Figure 1—figure supplement 1C*), a sensor for the SM-sequestered pool of PM cholesterol (*Figure 1—figure supplement 1A*; *Endapally et al., 2019a*; *Johnson et al., 2019*). In line with the immunoblot analysis in *Figure 1D & E*, microscopy imaging with fluorescent versions of ALOD4 (fALOD4-Neon) and OlyA (fOlyA-647) also showed that oxysterols D, G, H, I, and J eliminated accessible cholesterol, but not SM-sequestered cholesterol, from the PMs of CHO-K1 cells (*Figure 1—figure supplement 2*).

We followed up with dose curve analysis of the effect of each oxysterol on ALOD4 binding. At the highest concentrations tested, oxysterols D, G, H, I, and J reduced ALOD4 binding by more than 65%, whereas treatment with oxysterols A, B, C, E, F, or K showed no such reduction (*Figure 1—figure supplement 3A*). Detailed time course analyses for the five oxysterols that markedly reduced ALOD4 binding (D, G, H, I, and J) showed that oxysterols D, I, and J reduced binding by more than 50% within 1 hr, whereas oxysterols G and H required 4 hr for a similar reduction (*Figure 1—figure supplement 3B*). Together, these data indicate that selective depletion of accessible PM cholesterol within 4 hr is triggered by oxysterols harboring hydroxyl groups on the iso-octyl side chain at carbons 20 (*D*), 24 (*G*, *H*), 25 (*I*), or 27 (*J*).

To support our interpretation that the reduction of ALOD4 binding by certain oxysterols indicated depletion of accessible PM cholesterol, we carried out flow cytometry analysis of fALOD4-Neon binding to rabbit red blood cells (RBCs), which have an outer limiting membrane with high levels of accessible cholesterol but no internal membranes or mechanisms to synthesize, modify, or internalize cholesterol (*Chakrabarti et al., 2017*). As shown in *Figure 1—figure supplement 4A and B*, we

observed no reduction in fALOD4-Neon binding to RBCs that were treated for 4 hr with the panel of oxysterols (*Figure 1C*), whereas binding was abolished by treatment with hydroxypropyl β-cyclodextrin (HPCD), a reagent that extracts cholesterol from membranes (*Ohtani et al., 1989*). These results suggest that reduction of ALOD4 binding to the PMs of CHO-K1 cells by oxysterols *D*, *G*, *H*, *I*, and *J* is not due to effects on the lipid bilayer or on ALOD4, but rather due to their effects on cholesterol homeostatic pathways.

## Stimulation of ACAT activity triggers rapid depletion of accessible cholesterol from PMs

Oxysterols lower cellular cholesterol by targeting multiple cholesterol homeostatic systems, including ACAT, SREBPs, and LXRs. Having established which oxysterols rapidly deplete accessible PM cholesterol (*Figure 1D*), we sought to determine if one or more of these cholesterol homeostatic systems are regulated by the same oxysterols. We first tested the oxysterol specificity for activation of ACAT's enzymatic activity using a well-established cellular assay that measures ACAT's ability to attach [$^{14}$C] oleate, a radiolabeled fatty acid, to cholesterol to generate cholesteryl [$^{14}$C]oleate (*Goldstein et al., 1983*). In the absence of oxysterol treatment, ACAT activity in CHO-K1 cells was low (*Figure 1F*, *column 1*). Oxysterols *D*, *G*, *H*, *I*, and *J* stimulated a more than six-fold increase in ACAT activity, whereas oxysterols *A*, *B*, *C*, *E*, *F*, and *K* did not show a similar stimulatory effect (*Figure 1F*). This specificity for activation of ACAT exactly matched that for the depletion of accessible cholesterol from PMs (*Figure 1D*).

We then assessed the oxysterol specificity for inactivation of SREBPs with an assay that we routinely use to monitor the status of SREBP2, one of the three isoforms of SREBP (*Radhakrishnan et al., 2007*; *Horton et al., 2002*). In this experiment, CHO-K1 cells were first depleted of cholesterol by treatment with HPCD, which triggered the conversion of SREBP2 from its inactive precursor form (*P*) to its active cleaved nuclear form (*N*) (*Figure 1G*, lane 1). When the cholesterol-depleted cells were then treated with each of the oxysterols for 4 hr, we observed that oxysterols *B*, *D*, *E*, *G*, *H*, *I*, *J*, and *K* suppressed SREBP2 cleavage whereas oxysterols *A*, *C*, and *F* had a reduced effect (*Figure 1G*, lanes 2–12, and *Figure 1—figure supplement 5*). This specificity profile did not match that for depletion of accessible cholesterol from PMs (see summary table in *Figure 1H*). In particular, oxysterols *B*, *E*, and *K* suppressed SREBP2 cleavage but were unable to reduce ALOD4 binding (*Figure 1D*), suggesting that SREBP2 suppression does not drive the rapid depletion of accessible cholesterol from PMs.

Lastly, we evaluated the extensive previous literature that has addressed the oxysterol specificity for activating LXR transcription factors (*Janowski et al., 1996*; *Lehmann et al., 1997*; *Janowski et al., 1999*; *Ma et al., 2008*; *Berrodin et al., 2010*; *Nury et al., 2013*). These studies showed that oxysterols *A*, *B*, *E*, *G*, *H*, *I*, and *J* activate LXRs, whereas oxysterols *F* and *K* had no such activating effect (see summary table in *Figure 1H*). Similar to results with SREBP2, the oxysterol specificity for LXR activation also did not match that for reduction in ALOD4 binding.

## Disruption of ACAT activity abolishes the ability of oxysterols to rapidly deplete accessible cholesterol from PMs

The above oxysterol specificity analysis suggested that ACAT, but not SREBPs or LXRs, triggers the rapid depletion of accessible PM cholesterol. To test this hypothesis, we generated a CHO-K1 cell line that lacks ACAT enzymatic activity. Mammalian cells contain two isoforms of ACAT, ACAT1 and ACAT2, however previous studies of CHO cells have shown that ~99% of their ACAT activity arises from the ACAT1 isoform (*Cadigan et al., 1988*; *Chang et al., 1995*). Therefore, we used CRISPR-Cas9 genome editing to disrupt the gene encoding ACAT1 (*Soat1*) in CHO-K1 cells (*Figure 2—figure supplement 1A*). The resulting ACAT1-deficient cells, hereafter designated as ACAT1 knockout (KO) cells, had undetectable levels of ACAT1 protein (*Figure 2—figure supplement 1B*). Moreover, ACAT enzymatic activity, as judged by cholesteryl[$^{14}$C]oleate formation, was stimulated by 25HC, one of the five oxysterols that triggered ACAT activity (*Figure 1F*), to an expected degree in wild-type (WT) cells but not in ACAT1 KO cells (*Figure 2—figure supplement 1D*, *first two groups*), further indicating that the ACAT1 KO cells lacked ACAT activity. To ensure that these results were due to loss of ACAT activity, we stably expressed in the ACAT1 KO cells either wild-type human ACAT1 or a mutant version harboring a point mutation (H460A) that abolishes ACAT1's enzymatic activity, and the resulting cell lines were designated as ACAT1 KO; hACAT1(WT) and ACAT1 KO; hACAT1(H460A), respectively.

Compared to the ACAT1 KO cells, these two new cell lines contained high levels of ACAT1 protein (*Figure 2—figure supplement 1C*). Moreover, stimulation of ACAT enzymatic activity by 25HC was restored by the stable expression of hACAT1(WT), but not hACAT1(H460A) (*Figure 2—figure supplement 1D last two groups*). Thus, these four cell lines provide a framework to study the specific role of ACAT activity in determining accessible cholesterol levels in PMs.

We then measured the effects of 25HC on accessible cholesterol in the PMs of these four cell lines (*Figure 2* and *Figure 2—figure supplement 2*). As expected, treatment of WT cells with 25HC for 4 hr depleted PM accessible cholesterol in a dose-dependent manner (*Figure 2A*, *lanes 1–6, first panel*). In contrast, no such depletion was observed in ACAT1 KO cells (*Figure 2A*, *lanes 1–6, third panel*). Restoration of ACAT activity by stable expression of hACAT1(WT) reestablished the ability of 25HC to deplete accessible PM cholesterol, whereas 25HC had no effect in cells stably expressing an inactive mutant of hACAT1 (H460A) (*Figure 2A*, *lanes 1–6, fifth and seventh panels*). When each of these four cell lines were depleted of sterols and then treated with 25HC, we observed similar dose dependences for suppression of SREBP2 processing (*Figure 2—figure supplement 3*), indicating that 25HC entered the cells and reached their ER membranes to bind Insigs and prevent SREBP2 transport. Thus, the inability of 25HC to deplete accessible PM cholesterol in ACAT1 KO cells is not simply due to a defect in cellular entry of 25HC. In all four cell lines, 4HC did not diminish PM accessible cholesterol (*Figure 2A*, *lane 7, all panels*), consistent with this oxysterol's inability to stimulate ACAT activity (*Figure 1F, oxysterol A*). Similar to the results with 25HC, the four other oxysterols that depleted accessible PM cholesterol in WT cells (*Figure 1D, oxysterols D, G, H, and J*) were also unable to do so in the ACAT1 KO cells, further underscoring the link between ACAT and accessible PM cholesterol (*Figure 2—figure supplement 4*).

We also used fluorescence microscopy to analyze the effects of 25HC on the distribution of accessible and SM-sequestered pools of cholesterol in these cell lines. Consistent with immunoblot analysis (*Figure 2A*), we observed that 25HC treatment for 4 hr eliminated binding of fALOD4-Neon to WT cells, and this effect was abolished in ACAT1 KO cells (*Figure 2B*, *top two rows*). Likewise, expression of hACAT1(WT), but not hACAT1(H460A), in the ACAT1 KO cells restored 25HC's ability to deplete fALOD4-Neon binding (*Figure 2B*, *third and fourth rows*). In all four cell lines, binding of fOlyA-647 was unchanged by 25HC treatment (*Figure 2B*), further highlighting that over a period of 4 hr, 25HC specifically depletes the accessible pool of PM cholesterol in an ACAT-mediated manner.

To confirm that ACAT is the primary target of 25HC for rapid depletion of accessible PM cholesterol, we examined two previously described cell lines, one lacking all three isoforms of SREBPs (owing to a deficiency of Scap, which stabilizes SREBPs) (*Sakai et al., 1996*; *Rawson et al., 1999*) and the other lacking both isoforms of LXRs (*Abrams et al., 2020a*). Compared to WT cells, Scap-deficient cells (that have no detectable SREBPs) and LXR-deficient cells both had lower levels of accessible cholesterol on their PMs (*Figure 2C and D*, *left panels*, *lane 1*), and this pool of cholesterol was completely depleted by treatment with 25HC for 4 hr (*lanes 2–6*). Treatment with 4HC had no effect on accessible cholesterol levels in either Scap- or LXR-deficient cells (*Figure 2C and D*, *left panels*, *lane 7*). Also, levels of SM-sequestered cholesterol in these cells were unchanged after treatment with either 25HC or 4HC (*Figure 2C and D*, *right panels*, *lanes 1–7*).

Combined, the above sets of studies show that the rapid (1–4 hr) removal of accessible cholesterol from PMs by 25HC is due to its activation of ACAT and not due to its modulation of the SREBP or LXR pathways.

## A model for rapid depletion of accessible cholesterol from PMs by ER-localized ACAT

The concentration of cholesterol in the PM (~40–50 mole% of total lipids) is almost 10-fold higher than that in the ER membrane (~5 mole% of total lipids; *Lange et al., 1989*; *Das et al., 2013*; *Radhakrishnan et al., 2008*). How then can activation of ACAT, a cholesterol-modifying enzyme in the ER membrane, rapidly alter the levels of the accessible cholesterol pool in the PM? Previous studies have suggested that even though the total cholesterol concentrations in the PM and ER are quite different, the concentration of accessible cholesterol in the two membranes may be similar (*Lange et al., 2004*; *Wattenberg and Silbert, 1983*; *Radhakrishnan and McConnell, 2000*). The equivalence in accessible cholesterol levels between the PM and ER is maintained by rapid bi-directional transport of cholesterol between the two membranes (*Lange et al., 1993*; *Mesmin and Maxfield, 2009*).

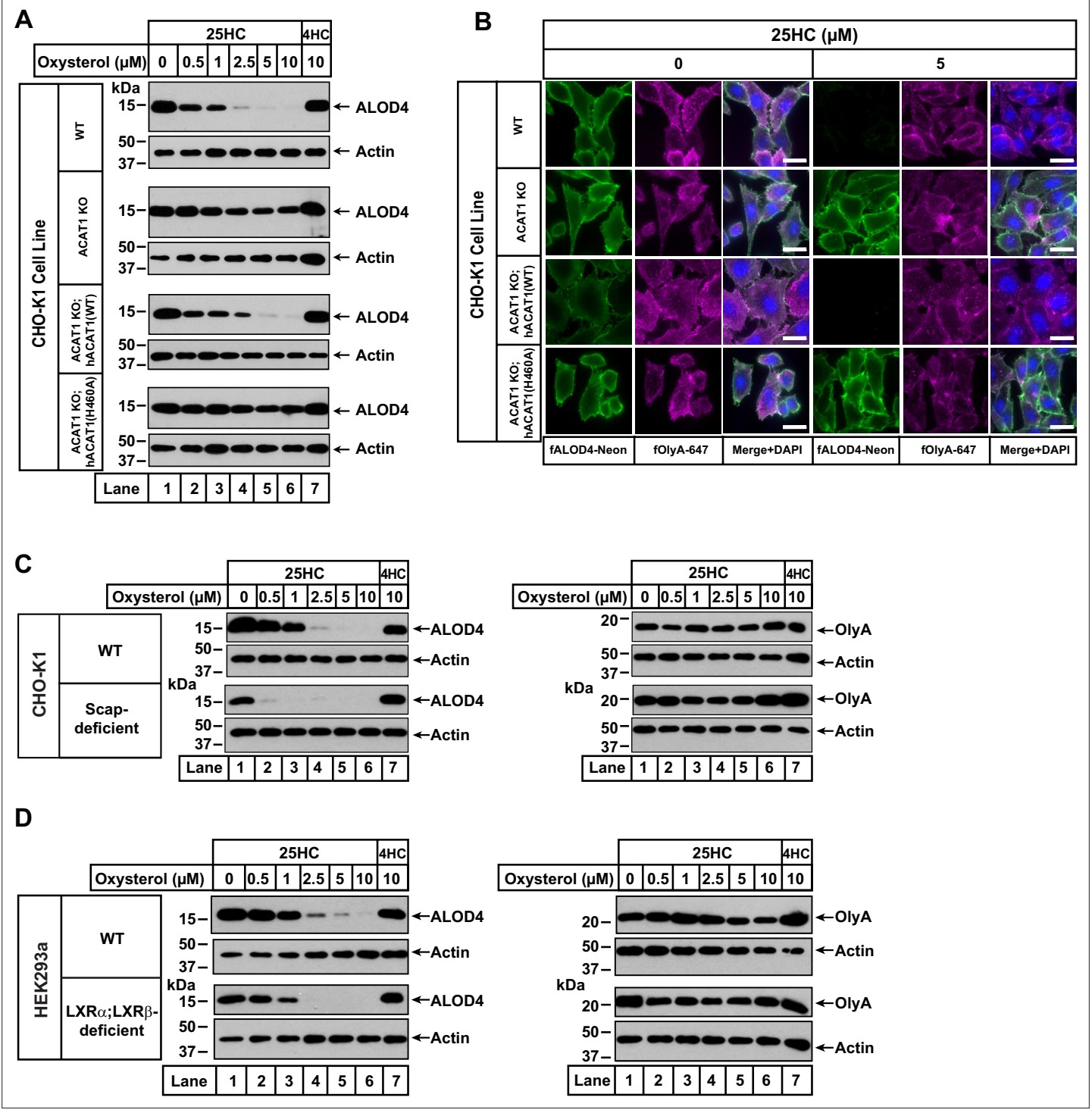

**Figure 2.** 25HC fails to trigger rapid depletion of accessible cholesterol from PMs of ACAT-deficient cells. (**A**) Immunoblot analysis of ALOD4 binding. On day 0, the indicated versions of CHO-K1 cells were set up in medium B at a density of $6\times10^4$ cells per well of a 48-well plate. On day 1, media was removed, cells were washed twice with 500 µl of PBS followed by addition of 200 µl of medium B supplemented with the indicated concentrations of either 25HC or 4HC. After incubation at 37 °C for 4 hr, media was removed and replaced with 200 µl of medium B supplemented with 3 µM His$_6$-Flag-ALOD4. After incubation at 37 °C for 30 min, cells were washed twice with 500 µl of PBS, harvested, and equal aliquots of cell lysates were subjected to immunoblot analysis as described in *Materials and methods*. Quantification of the immunoblots are shown in *Figure 2—figure supplement 2*. (**B**) Fluorescence microscopy analysis of ALOD4 and OlyA binding. On day 0, the indicated versions of CHO-K1 cells were set up in medium B at a density of $3\times10^4$ cells per well of an eight-well Lab-Tek II chambered #1.5 coverglass dish. On day 1, media was removed, cells were washed twice with

*Figure 2 continued on next page*

*Figure 2 continued*

500 µl of PBS followed by addition of 200 µl of medium B supplemented with the indicated concentration of 25HC. After incubation at 37 °C for 4 hr, media was removed and replaced with 200 µl of medium B supplemented with 3 µM of either fALOD4-Neon or fOlyA-647. After incubation at 37 °C for 30 min, media was removed, cells were washed twice with 500 µl of PBS, fixed, stained with DAPI, and imaged as described in *Materials and methods*. *Scale bar*, 25 µm. (**C, D**) Effects of 25HC on PM cholesterol pools in cells lacking Scap or LXR transcription factors. On day 0, the indicated cell lines were set up in either medium B (**C**) or medium D (**D**) at a density of 6x10⁴ cells per well of a 48-well plate. On day 1, media was removed, cells were washed twice with 500 µl of PBS followed by addition of 200 µl of media supplemented with the indicated concentrations of either 25HC (*lanes 1–6*) or 4HC (*lane 7*). After incubation at 37 °C for 4 hr, media was removed and replaced with 200 µl of media supplemented with 3 µM of either His₆-Flag-ALOD4 (*left panels*) or OlyA-His₆ (*right panels*). After incubation at 37 °C for 30 min, cells were washed twice with 500 µl of PBS, harvested, and equal aliquots of cell lysates were subjected to immunoblot analysis as described in *Materials and methods*. Quantification of the immunoblots are shown in *Figure 2— figure supplement 5*.

The online version of this article includes the following source data and figure supplement(s) for figure 2:

**Source data 1.** Western blots corresponding to *Figure 2A, C and D*.

**Figure supplement 1.** Characterization of a CHO-K1 cell line deficient in ACAT1.

**Figure supplement 1—source data 1.** Western blots corresponding to *Figure 2—figure supplement 1B, C*.

**Figure supplement 1—source data 2.** Data supporting *Figure 2—figure supplement 1D*.

**Figure supplement 2.** Quantification of immunoblot analysis shown in *Figure 2A*.

**Figure supplement 2—source data 1.** Data supporting *Figure 2—figure supplement 2*.

**Figure supplement 3.** Effects of 25HC on SREBP2 processing in ACAT-deficient cells.

**Figure supplement 3—source data 1.** Western blots corresponding to *Figure 2—figure supplement 3*.

**Figure supplement 4.** Oxysterols that activate ACAT fail to trigger rapid depletion of accessible cholesterol from PMs of ACAT-deficient cells.

**Figure supplement 4—source data 1.** Western blots corresponding to *Figure 2—figure supplement 4*.

**Figure supplement 5.** Quantification of immunoblot analysis shown in *Figure 2C and D*.

**Figure supplement 5—source data 1.** Data supporting *Figure 2—figure supplement 5A and B*.

Based on these previous insights, we propose that ACAT activation, and the resulting production of cholesteryl esters, would siphon off some of the accessible cholesterol from the ER, thus lowering levels of accessible cholesterol in the ER relative to the PM. Cholesterol transport from the PM to the ER would then rapidly restore the equivalence of accessible cholesterol between the two membranes. As a result, levels of accessible cholesterol in the PM would decline.

The above model provides a possible explanation for how ACAT activation by 25HC would lead to a depletion of accessible PM cholesterol in WT cells, but not in ACAT1 KO cells (*Figure 2A*). We further tested this model by activating ACAT not by 25HC but by lipoproteins, which have been shown to generate high ACAT activity (*Goldstein et al., 1983*). We then took advantage of the sensitivity of flow cytometry analysis to assess levels of accessible and SM-sequestered cholesterol in the PMs of WT and ACAT1 KO cells with fluorescent versions of ALOD4 and OlyA, respectively. When cells are grown under low cholesterol conditions (lipoprotein-deficient serum, LPDS), ACAT activity is low (*Goldstein et al., 1983*). Under these conditions, both WT and ACAT1 KO cells had the same low levels of accessible and SM-sequestered cholesterol in their PMs (*Figure 3A*). By comparison, when cells are grown under high cholesterol conditions (FCS), ACAT activity is high (*Goldstein et al., 1983*). In this case, the PMs of ACAT1 KO cells had significantly higher levels of accessible cholesterol as compared to WT cells (*Figure 3B*). This result is consistent with the model proposed above since the ACAT1 KO cells, unlike the WT cells, are unable to siphon away ER cholesterol, leading to higher levels of accessible cholesterol in the PMs of these cells. Importantly, there was no difference in levels of SM-sequestered cholesterol in the PMs of WT and ACAT1 KO cells (*Figure 3B*), highlighting the specific effect of the ACAT pathway on accessible PM cholesterol.

An assumption of our model is that rapid bi-directional transport of accessible cholesterol between the PM and the ER was not disrupted in ACAT1 KO cells. To test this notion, we carried out three different experiments in which the cholesterol content of the PM was varied and changes in ER cholesterol were monitored by assessing the molecular size of SREBP2, which is processed from its precursor form (*P*) to a cleaved nuclear form (*N*) in response to decreases in ER cholesterol (*Radhakrishnan et al., 2008*). First, we depleted cholesterol from the PMs of FCS-treated cells by incubation with HPCD, which extracts cholesterol from PMs. The reduction in PM cholesterol is rapidly compensated

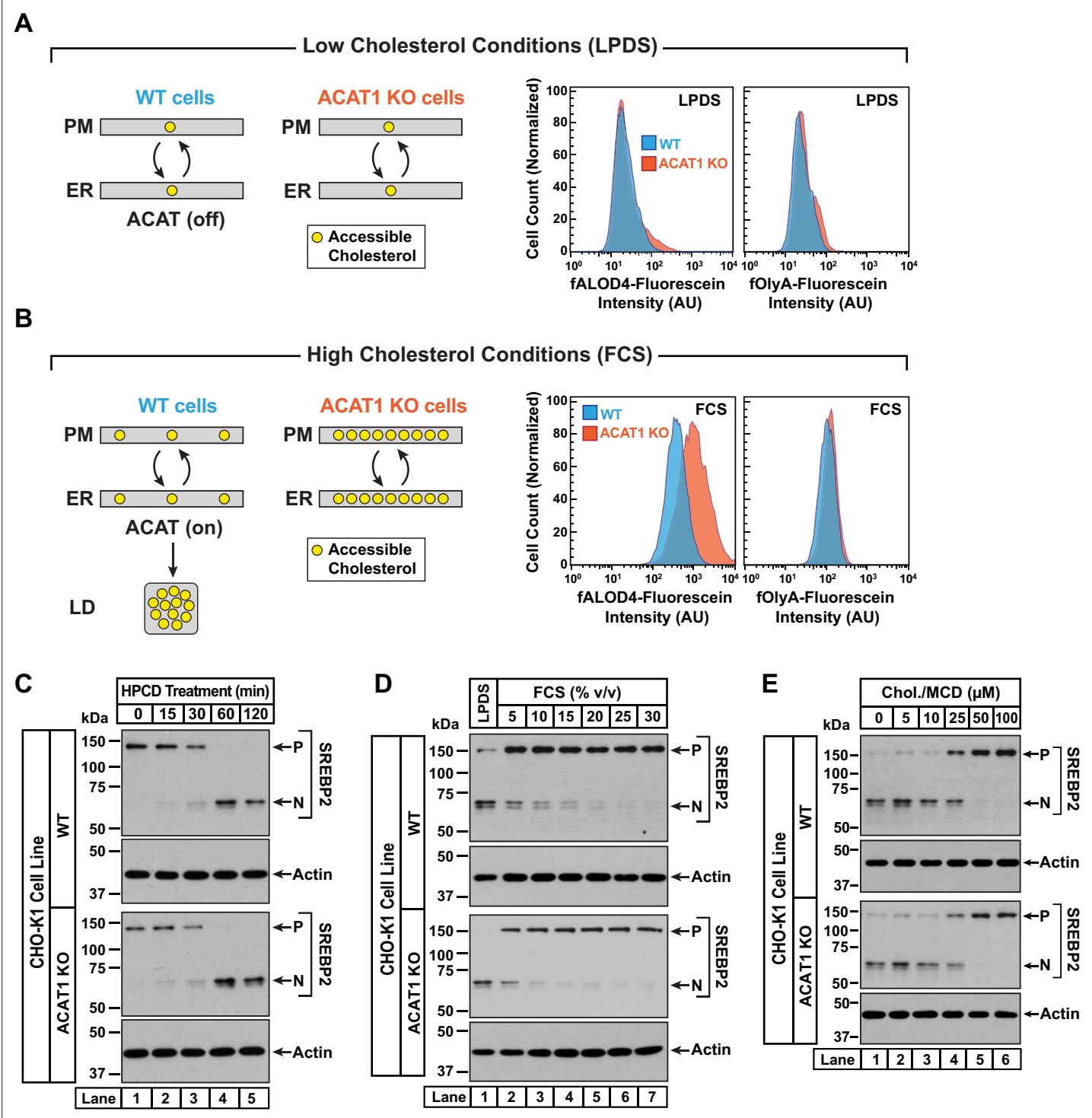

**Figure 3.** Intracellular cholesterol trafficking and steady state levels of PM accessible cholesterol in ACAT-deficient cells. On day 0, wild-type and ACAT1-deficient CHO-K1 cells were set up in medium B at a density of 5x10⁴ cells per well of a 24-well plate (**A, B**) or 6x10⁴ cells per well of a 48-well plate (**C - E**). (**A, B**) Steady state levels of accessible cholesterol in PMs. On day 1, media was removed, cells were washed twice with 500 μl of PBS followed by addition of 500 μl of medium A supplemented with 5% (v/v) of either LPDS (**A**) or FCS (**B**). On day 2, media was removed, cells were washed twice with 500 μl of PBS followed by addition of 200 μl of medium A containing 5% (v/v) of the indicated serum along with 0.5 μM of either fALOD4-fluorescein or fOlyA-fluorescein. After incubation at 37 °C for 30 min, media was removed, cells were washed twice with 500 μl of PBS, and subjected to flow cytometry as described in *Materials and methods. LD*, lipid droplets. (**C**) Cholesterol depletion by HPCD. On day 1, media was removed, cells were washed twice with 500 μl of PBS followed by addition of 200 μl of medium B supplemented with 2% (w/v) HPCD. (**D**) Cholesterol repletion

*Figure 3 continued on next page*

*Figure 3 continued*

by lipoproteins. On day 1, media was removed, cells were washed twice with 500 µl of PBS followed by addition of 500 µl of cholesterol-depleting medium C. On day 2, media was removed, cells were washed twice with 500 µl of PBS followed by addition of 200 µl of medium A containing either 5% (v/v) lipoprotein-deficient serum (*lane 1*) or the indicated concentrations of lipoprotein-rich FCS (*lanes 2–7*). (**E**) Cholesterol repletion by cholesterol/cyclodextrin complexes. On day 1, media was removed, cells were washed twice with 500 µl of PBS followed by addition of 200 µl of medium C supplemented with 1% (w/v) HPCD. After incubation at 37 °C for 1 hr, media was removed, cells were washed twice with 500 µl of PBS followed by addition of 200 µl of medium C supplemented with the indicated concentrations of cholesterol/MCD complexes. (**C – E**) After incubation at 37 °C for the indicated times (**C**), 5 hr (**D**), or 3 hr (**E**), media was removed, cells were washed twice with 500 µl of PBS, harvested, and equal aliquots of cell lysates were subjected to immunoblot analysis as described in *Materials and methods*. P, precursor form of SREBP2; N, cleaved nuclear form of SREBP2; *Chol.*, cholesterol. Quantification of the immunoblots are shown in *Figure 3—figure supplement 1*.

The online version of this article includes the following source data and figure supplement(s) for figure 3:

**Source data 1.** Western blots corresponding to *Figure 3C, D and E*.

**Figure supplement 1.** Quantification of immunoblot analysis shown in *Figure 3C, D and E*.

**Figure supplement 1—source data 1.** Data supporting *Figure 3—figure supplement 1A, B, C*.

for by drawing cholesterol from the ER, leading to a reduction in ER cholesterol levels and an increase in SREBP2 cleavage. In both WT and ACAT1 KO cells, HPCD treatment reduced ER cholesterol levels with similar time dependences (*Figure 3C* and *Figure 3—figure supplement 1A*), indicating that the pathways that transport cholesterol from ER to PM are not affected in the ACAT1 KO cells.

Second, we depleted both cell lines of cholesterol by incubation overnight in the presence of lipoprotein-deficient serum along with compactin, an inhibitor of cholesterol synthesis (*Brown et al., 1978*). This treatment lowered ER cholesterol to a similar degree in both cell lines, as judged by similar levels of nuclear SREBP2 (*Figure 3D*, *lane 1*, *first and third panels*). We then added back increasing concentrations of lipoprotein-rich FCS and observed a similar dose-dependent increase in ER cholesterol in both cell lines, as judged by reduction of nuclear SREBP2 (*Figure 3D*, *lanes 2–7*, and *Figure 3—figure supplement 1B*). Third, we depleted cells of cholesterol by treatment with HPCD and then added back increasing concentrations of complexes of cholesterol with methyl-β-cyclodextrin (MCD), which deliver cholesterol to the PMs. Analysis of SREBP2 cleavage in both cell lines showed similar reduction in ER cholesterol levels upon depletion (*Figure 3E*, *lane 1*, *first and third panels*), and a similar dose-dependent increase in ER cholesterol levels upon replenishment with cholesterol/MCD (*Figure 3E*, *lanes 2–6*, and *Figure 3—figure supplement 1C*). The results of *Figure 3D and E* suggest that the pathways transporting lipoprotein-derived or exogenously added cholesterol from PM to ER are not affected in the ACAT1 KO cells.

Taken together, the results in *Figure 3* provide a plausible mechanism for how 25HC and other ACAT-activating oxysterols could rapidly alter the levels of accessible cholesterol in the PMs of cells whose overall cholesterol content varies over a wide range.

## 25HC-triggered depletion of accessible PM cholesterol persists long after 25HC removal through suppression of SREBPs

We next wondered whether the rapid depletion of accessible PM cholesterol after a short exposure to 25HC would be sustained over longer periods. To explore this possibility, we first treated CHO-K1 cells with 25HC for a short period of 4 hr, which depleted accessible cholesterol from the PMs in an expected manner (*Figure 4A*, *first panel*, *lanes 1, 2*). We then removed the 25HC and examined accessible PM cholesterol levels over time. Despite the lack of 25HC in the extracellular media and cells being grown in cholesterol-rich FCS, we detected no replenishment of accessible PM cholesterol even after 22 hr (*Figure 4A*, *first panel*, *lanes 3–8*). In line with our model of equivalence of accessible cholesterol levels in the PM and ER (*Figure 3A*), we expected that ER accessible cholesterol levels would also be low 22 hr after 25HC removal, which would be reflected in increased cleavage of SREBP2 to its nuclear form. Surprisingly, we detected no processing of SREBP2 (*Figure 4A*, *third panel*). Mass spectrometric analysis of intracellular 25HC levels (*Figure 4A*, *bottom panel*) showed that 25HC was barely detectable (~0.015 ng/µg of cellular protein) in untreated cells (*lane 1*) and rose to 4 ng/µg of cellular protein after 25HC treatment for 4 hr (*lane 2*). After removal of 25HC from the extracellular medium, the intracellular 25HC declined over the time course of 22 hr to 0.35 ng/µg of cellular protein (*lanes 3–8*), a level that was 23-times higher than that in untreated cells and sufficient to suppress SREBP2 cleavage (*Figure 4A*, *lane 8*).

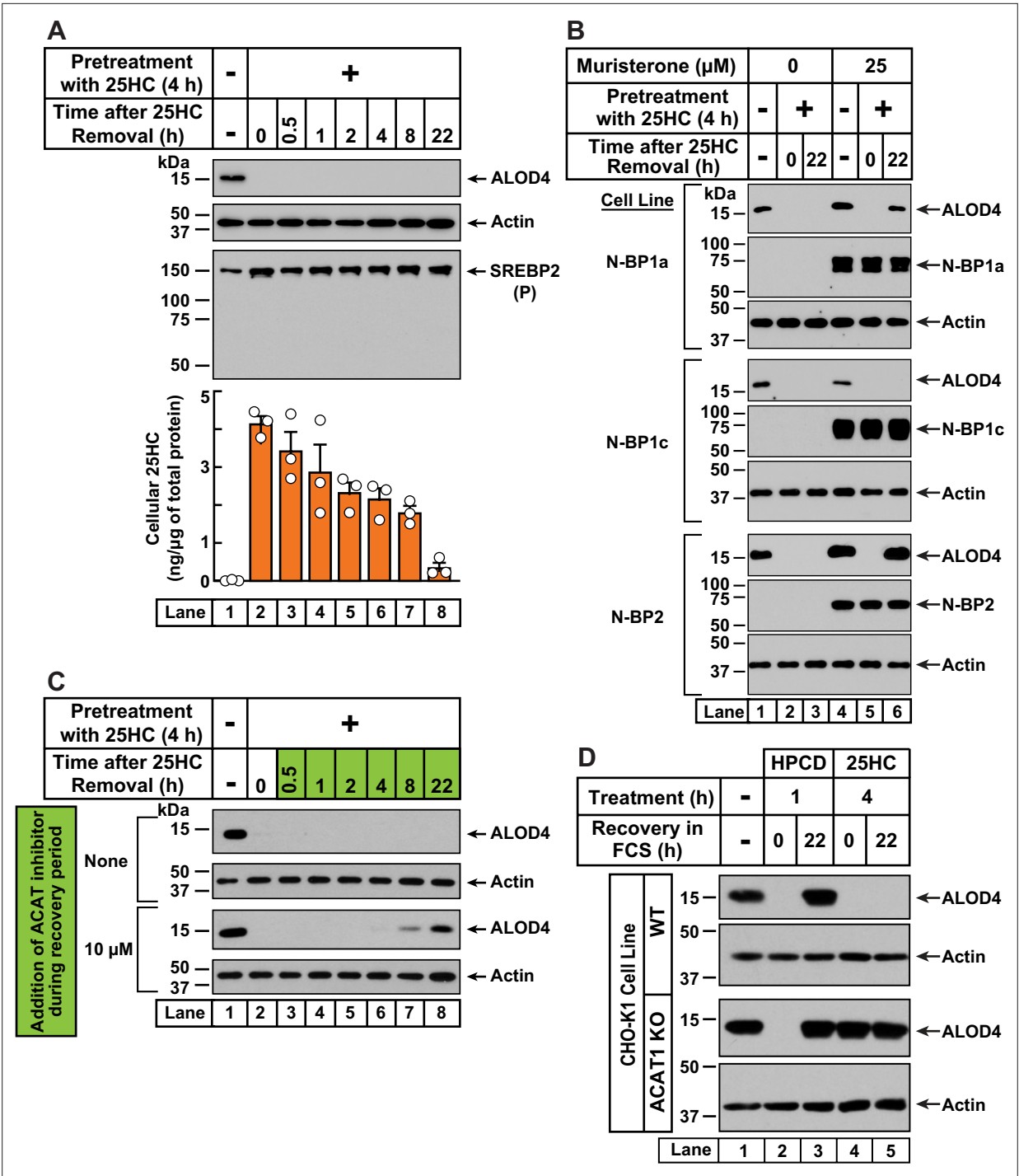

**Figure 4.** 25HC-triggered depletion of accessible cholesterol from PMs persists for long times through suppression of SREBP-mediated cholesterol synthesis and uptake, and continued activation of ACAT. (**A**) Retention of 25HC in cells prevents replenishment of accessible cholesterol on PMs after it has been depleted by 25HC. On day 0, CHO-K1 cells were set up in medium B in eight 24-well plates at a density of 1.5x10⁵ cells per well. On day 1, media was removed from seven of the eight plates, cells were washed twice with 1 ml of PBS followed by addition of 1 ml of medium B supplemented with 5 μM of 25HC (*lanes 2–8*). After incubation at 37 °C for 4 hr, media was removed, cells were washed twice with 1 ml of PBS followed by addition of 1 ml of medium B. Each of the seven plates was incubated for the indicated times, after which media was removed from 10 wells and cells were washed twice with 1 ml of PBS. Then, 100 μl of PBS was added to each well and the cells were scraped and pooled for further analysis. The eighth plate (*lane 1*) was not subjected to any treatment and was processed for analysis as above. For each plate, an aliquot of the pooled cells (20% of total) was used to determine protein concentration with a BCA protein assay kit and the remainder (80% of total) was subjected to mass spectrometry analysis for free 25HC as described in *Materials and methods*. An 11th well from each plate was subjected to immunoblot analysis of His₆-Flag-ALOD4 binding (*top

*Figure 4 continued on next page*

*Figure 4 continued*

panel). These samples were also immunoblotted for SREBP2 (*third panel*). P, precursor form of SREBP2; N, cleaved nuclear form of SREBP2. (**B**) Nuclear SREBPs counteract 25HC-mediated depletion of accessible cholesterol. On day 0, Site-2 protease-deficient CHO-K1 cells inducibly expressing the nuclear transcription factor domains (N–BP) of the indicated isoforms of SREBP were set up in medium B at a density of $5\times10^4$ cells per well of a 48-well plate. On day 1, media was removed, cells were washed twice with 500 µl of PBS followed by addition of 200 µl of medium B supplemented with the indicated concentrations of muristerone A. On day 2, media was removed, cells were washed twice with 500 µl of PBS followed by addition of 200 µl of medium B supplemented with the indicated concentration of muristerone along with 5 µM 25HC (*lanes 2, 3, 5, 6*). After incubation at 37 °C for 4 hr, media was removed, cells were washed twice with 500 µl of PBS followed by addition of 200 µl of medium B. After incubation for the indicated times, media was removed from these 25HC-treated wells as well as from two wells that were not subjected to any of the above treatments (*lanes 1, 4*). The media was replaced with 200 µl of medium B supplemented with 3 µM His$_6$-Flag-ALOD4. After incubation at 37 °C for 30 min, cells were washed twice with 500 µl of PBS, harvested, and equal aliquots of cell lysates were subjected to immunoblot analysis as described in *Materials and methods*. Quantification of the immunoblots are shown in *Figure 4—figure supplement 2A*. (**C**) Inhibition of ACAT activity allows partial replenishment of accessible cholesterol on PMs after it has been depleted by 25HC. On day 0, CHO-K1 cells were set up in medium B at a density of $1.5\times10^5$ cells per well of a 24-well plate. On day 1, media was removed, cells were washed twice with 1 ml of PBS, followed by addition of 1 ml of medium B supplemented with 5 µM 25HC. After incubation at 37 °C for 4 hr, media was removed, cells were washed twice with 1 ml of PBS, followed by addition of 1 ml of medium B without or with 10 µM of SZ58-035. After incubation for the indicated times, media was removed and replaced with 200 µl of medium B supplemented with 3 µM His$_6$-Flag-ALOD4. After incubation at 37 °C for 30 min, cells were washed twice with 1 ml of PBS, harvested, and equal aliquots of cell lysates were subjected to immunoblot analysis as described in *Materials and methods*. Quantification of the immunoblots are shown in *Figure 4—figure supplement 2B*. (**D**) PM accessible cholesterol in ACAT-deficient cells. On day 0, the indicated versions of CHO-K1 cells were set up in medium B at a density of $1.5\times10^5$ cells per well of a 24-well plate. On day 1, media was removed, cells were washed twice with 500 µl of PBS followed by addition of 300 µl of medium B supplemented with either 1% (w/v) HPCD (*lanes 2, 3*) or 5 µM 25HC (*lanes 4, 5*). After incubation for either 1 hr (*lanes 2, 3*) or 4 hr (*lanes 4, 5*), media was removed, cells were washed twice with PBS followed by addition of 1 ml of medium B. After incubation for the indicated times, media was removed from these treated wells as well as from a well that was not subjected to any of the above treatments (*lane 1*). The media was replaced with 200 µl of medium B supplemented with 3 µM His$_6$-Flag-ALOD4. After incubation at 37 °C for 30 min, cells were washed twice with 1 ml of PBS, harvested, and equal aliquots of cell lysates were subjected to immunoblot analysis as described in *Materials and methods*. Quantification of the immunoblots are shown in *Figure 4—figure supplement 2C*.

The online version of this article includes the following source data and figure supplement(s) for figure 4:

**Source data 1.** Western blots corresponding to *Figure 4*.

**Source data 2.** Data supporting *Figure 4A* and *Figure 4—figure supplement 2*.

**Figure supplement 1.** Recovery of accessible cholesterol on PMs after depletion by HPCD.

**Figure supplement 1—source data 1.** Western blots corresponding to *Figure 4—figure supplement 1*.

**Figure supplement 2.** Quantification of immunoblot analysis shown in *Figure 4B, C and D*.

In contrast to the sustained action of 25HC, when accessible PM cholesterol was depleted by treatment with HPCD (1 hr), its levels were replenished within 4 hr of HPCD removal (*Figure 4—figure supplement 1*, *first panel*). Moreover, HPCD depletion triggered SREBP2 cleavage in an expected fashion (*Figure 4—figure supplement 1*, *second panel*, *lanes 1, 2*), and this cleavage was suppressed at the same time as an increase in PM accessible cholesterol was detected (*Figure 4—figure supplement 1*, *second panel, lanes 3–8*). These data further support the notion that 25HC's long-term effects arise due to its suppression of SREBP cleavage. If this was the case, we would expect 25HC's effects to be overcome by expression of the cleaved nuclear forms of SREBPs. We tested this possibility in previously developed cell lines where expression of the cleaved nuclear forms of each of the three isoforms of SREBPs – SREBP1a, SREBP1c, and SREBP2 – can be induced by addition of the small molecule muristerone (*Pai et al., 1998*; *Figure 4B* and *Figure 4—figure supplement 2A*). In the absence of muristerone, these cells had detectable levels of accessible cholesterol in their PMs (*Figure 4B*, *lane 1*). 25HC treatment depleted this pool and no replenishment was observed 22 hr after the 25HC was removed (*Figure 4B*, *lanes 2, 3*). We then treated the cells with muristerone to induce the expression of the nuclear forms of each of the three SREBPs (*Figure 4B*, *lanes 4–6*). Under these conditions, the accessible cholesterol in the PMs of these cells was also depleted by 25HC (*Figure 4B*, *lanes 4, 5*). However, when the 25HC was removed, we now detected replenishment of accessible cholesterol on the PMs of cells expressing the nuclear forms of SREBP1a and SREBP2, but not SREBP1c (*Figure 4B*, *lane 6*). This isoform specificity for restoring accessible PM cholesterol is consistent with the previous observation that the nuclear forms of SREBP1a and SREBP2, but not SREBP1c, increase cholesterol synthesis and induce expression of the LDL receptor allowing for uptake of cholesterol from the LDL in FCS (*Pai et al., 1998*).

While the results of *Figure 4B* point to an important role for SREBP suppression in mediating 25HC's long-term depletion of accessible PM cholesterol, we wondered whether continued activation of ACAT by the lingering 25HC could also play a role. To test this possibility, we treated CHO-K1 cells with 25HC for 4 hr, which depleted accessible cholesterol from the PM (*Figure 4C*, *first and third panels*, *lanes 1, 2*). We then removed the 25HC and added back media containing cholesterol-rich FCS without and with SZ58-035, a small molecule that potently inhibits ACAT enzymatic activity (*Ross et al., 1984*). In the absence of SZ58-035, we observed no replenishment of accessible PM cholesterol (*Figure 4C*, *first panel*, *lanes 3–8*, and *Figure 4—figure supplement 2B*). In contrast, in the presence of SZ58-035, we began to detect accessible cholesterol on PMs after 8 hr and ~60% of the initial level of accessible PM cholesterol was restored after 22 hr (*Figure 4C*, *third panel, lanes 3–8*, and *Figure 4—figure supplement 2B*).

Based on these data, we hypothesized that 25HC acts through two distinct, temporally resolved stages. First, a short pulse of 25HC activates ACAT, which siphons away ER cholesterol resulting in rapid depletion of accessible cholesterol from the PM. Second, 25HC's suppression of SREBP-mediated uptake and synthesis of cholesterol combined with continued activation of ACAT, results in sustained depletion of accessible cholesterol in the PM. We next asked whether initial ACAT activation was required for depletion of accessible cholesterol over long time periods, or whether this could be achieved simply through sustained SREBP inhibition by residual 25HC? To answer this question, we compared the responses of WT and ACAT1 KO cells to depletion of accessible PM cholesterol (*Figure 4D* and *Figure 4—figure supplement 2C*). As expected, HPCD treatment (1 hr) induced the depletion of accessible PM cholesterol in both cell lines (*Figure 4D*, *lanes 1, 2*), and this depletion was fully reversed 22 hr after HPCD removal (*lane 3*), indicating that this mode of cholesterol depletion occurred independently of ACAT activity. A different result was observed after 25HC treatment for 4 hr. In WT cells, 25HC depleted the accessible cholesterol as expected and no replenishment was observed after the 25HC was removed (*Figure 4D*, *first panel*, *lanes 4, 5*). However, in ACAT1 KO cells, 25HC did not deplete accessible PM cholesterol and there was no further change after the 25HC was removed (*Figure 4D*, *third panel*, *lanes 4, 5*). Thus, without the initial action of 25HC on ACAT, the lingering intracellular 25HC that inhibits new cholesterol synthesis and uptake through suppressing SREBP activation is not enough to reduce accessible PM cholesterol, even after 22 hr.

## ACAT activation by 25HC protects cells from pore formation by bacterial cytolysins

Having determined how ACAT-activating oxysterols trigger both rapid and sustained depletion of accessible cholesterol from the PM, we next sought to determine if these mechanisms are responsible for the previously reported immunological effects of 25HC. Recent studies have indicated that 25HC protects cells from pore-forming activities of bacterial cholesterol-dependent cytolysins (CDCs) that exploit accessible PM cholesterol (*Zhou et al., 2020*; *Ormsby et al., 2022*; *Ormsby et al., 2021*). We used ACAT1 KO CHO-K1 cells as a model to test whether ACAT activation is the primary mechanism for 25HC's protection from pore formation by two CDCs, Anthrolysin O (ALO) (*Mosser and Rest, 2006*; *Shannon et al., 2003*) and Perfringolysin O (PFO) (*Shimada et al., 2002*; *Tweten et al., 2015*). Dose curve analysis showed that much lower concentrations of ALO were required to form pores in ACAT1 KO cells compared to WT cells (*Figure 5—figure supplement 1A*, *left panel*), consistent with the higher amounts of accessible cholesterol in the PMs of ACAT1 KO cells (*Figure 3B*). Moreover, stable expression of hACAT1(WT), but not inactive hACAT1(H460A), in the ACAT1 KO cells rendered them less susceptible to pore formation by ALO (*Figure 5—figure supplement 1A*, *right panel*). Similar results were obtained with PFO, although the extents of the dose dependency shifts were less dramatic (*Figure 5—figure supplement 1B*).

Next, we treated each of these four cell lines with 25HC for 4 hr and then measured the ability of ALO and PFO to rapidly form pores. As expected, 25HC treatment of WT cells abolished pore formation by ALO and PFO in a dose-dependent manner (*Figure 5*, *first column*), consistent with 25HC's ability to deplete accessible cholesterol from the PMs of these cells (*Figure 2A*). In contrast, 25HC treatment had no effect on pore formation in ACAT1 KO cells (*Figure 5*, *second column*) due to its inability to deplete accessible cholesterol from the PMs of these cells (*Figure 2A*). Restoration of ACAT activity by stable expression of hACAT1(WT) reestablished the ability of 25HC to abolish pore formation, whereas inactive hACAT1(H460A) was unable to do so (*Figure 5*, *third and fourth*

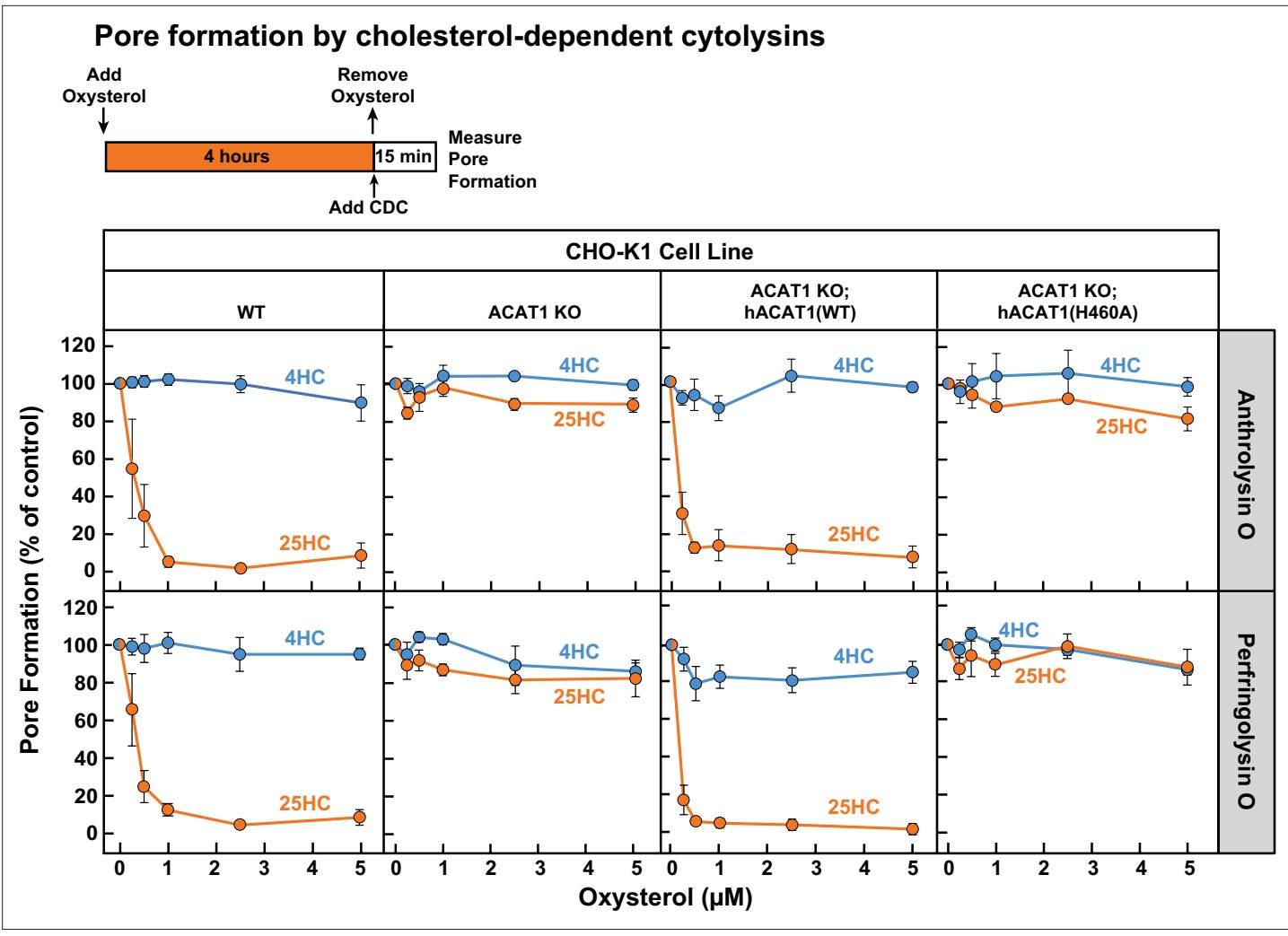

**Figure 5.** 25HC fails to protect ACAT-deficient cells from pore formation by bacterial cytolysins. On day 0, the indicated versions of CHO-K1 cells were set up in medium B at a density of $7.5\times10^4$ cells per well of a 48-well plate. On day 1, media was removed and replaced with 500 µl of medium B supplemented with the indicated concentrations of either 25HC or 4HC. After incubation at 37 °C for 4 hr, media was removed, cells were washed twice with 500 µl of HBSS followed by addition of 500 µl of HBSS containing either 100 pM of His$_6$-ALO(FL) (*top panel*) or 500 pM of His$_6$-PFO(FL) (*bottom panel*). After incubation at 37 °C for 15 min, media was removed and pore formation was assessed as described in *Materials and methods*. For each cell line, the extent of pore formation in the absence of oxysterol treatment was set to 100%, and all other values were normalized to this set-point.

The online version of this article includes the following source data and figure supplement(s) for figure 5:

**Source data 1.** Data supporting *Figure 5* and *Figure 5—figure supplement 1*.

**Figure supplement 1.** Susceptibility of wild-type and ACAT-deficient CHO-K1 cells to pore formation by bacterial cytolysins.

columns). In all four cell lines, 4HC had no effects on pore formation (*Figure 5*), consistent with this oxysterol's inability to deplete accessible cholesterol from PMs (*Figure 1D*). The above data obtained with ACAT1 KO CHO-K1 cells indicates that the protective effects of 25HC on macrophage lysis by CDCs primarily relies on ACAT activation.

## Anti-bacterial activity of 25HC requires ACAT activation

Unlike bacterial toxins that induce rapid lysis of host cells, bacterial pathogens such as *Listeria mono-cytogenes* can survive for long periods of time within host cells (*Portnoy et al., 2002*; *Radoshevich and Cossart, 2018*). We have previously shown that accessible cholesterol is required for *Listeria* to form membrane protrusions at cellular junctions and spread into neighboring cells, and that 25HC treatment for 18 hr blocks this spread (*Abrams et al., 2020a*; *Abrams et al., 2020b*). However, given that 25HC production by immune cells can be short-lived (*Bauman et al., 2009*; *Dennis et al., 2010*)

and accessible cholesterol depletion by 25HC occurs rapidly (*Figure 1A*), we asked if a short pulse of 25HC could suppress long-term infection by *Listeria*. The increased accessible PM cholesterol in ACAT1 KO cells did not affect invasion or cell-to-cell spread of *Listeria* as similar levels of infection were observed in both WT and ACAT1 KO cells after 22 hr (*Figure 6—figure supplement 1*). Moreover, infection levels were similar in the ACAT1 KO cells that stably expressed either hACAT1(WT) or inactive hACAT1(H460A) (*Figure 6—figure supplement 1*). In contrast to these results, short-term incubation of WT cells with 25HC prior to infection (followed by removal of the oxysterol) reduced *Listeria* infection in a dose dependent manner (*Figure 6A*, *first column*), but this treatment had no effect on infection of ACAT1 KO cells (*Figure 6A*, *second column*). The antibacterial activity of 25HC was restored by expression of hACAT1(WT), whereas inactive hACAT1(H460A) did not reduce infection (*Figure 6A*, *third and fourth columns*). In all four cell lines, 4HC had no effects on *Listeria* infection (*Figure 6A*), consistent with this oxysterol's inability to deplete accessible cholesterol from PMs (*Figure 1D*).

Previously, we showed that genetic ablation of cholesterol 25-hydroxylase (*Ch25h*), the enzyme that produces 25HC, in mice enhanced the transmission of *Listeria* from the gut to the spleen, and that delivery of exogenous 25HC suppressed this systemic transmission (*Abrams et al., 2020a*). To determine if the protective effects of 25HC observed in these studies were dependent on ACAT1 activation, we assessed the infectivity of ACAT1 global knock-out mice (*Meiner et al., 1996*). To obtain robust mucosal infection, we used a murinized *Listeria monocytogenes* EGD strain carrying a mutation in Internalin A that enhances its binding affinity for mouse E-cadherin (*Lm*-InlAᵐ) (*Wollert et al., 2007*). Consistent with our previous studies, transmission of *Lm*-InlAᵐ to liver and spleen tissues was significantly reduced in wild-type mice treated with 25HC (*Figure 6B, top row*). However, 25HC was unable to protect ACAT1 KO mice from *Lm*-InlAᵐ infection (*Figure 6B, bottom row*). In contrast to what we observed in liver and spleen tissues, we observed no difference in *Lm*-InlAᵐ levels in the caecal tissues of wild-type and ACAT1 KO mice upon 25HC treatment (*Figure 6B*), indicating that 25HC protects the organism from systemic transmission and not initial bacterial invasion of the gut epithelium. These data are consistent with those obtained in cultured cells and provide evidence that 25HC elicits long-term protective effects in vivo through activation of ACAT.

## 25HC does not protect ACAT-deficient cells from infection by two model viruses

The above studies show that ACAT activation by 25HC is a defense mechanism used by cells to combat bacteria and their toxins. 25HC has also been shown to suppress highly pathogenic viruses including human immunodeficiency virus (HIV), Ebola virus, Nipah virus, Rift Valley fever virus, Zika virus (ZIKV), and SARS-CoV-2, the causative agent of COVID-19 (*Wang et al., 2020*; *Liu et al., 2013*; *Li et al., 2017*; *Zu et al., 2020*; *Zang et al., 2020*). We asked if the antiviral activity of 25HC was also due to the long-lasting depletion of accessible PM cholesterol initiated by ACAT activation. We focused on infection by ZIKV and the model human coronavirus (hCoV-OC43) in Huh7.5 cells, a human hepatoma cell line that is permissive to viral infection due to defects in innate immune signaling (*Blight et al., 2002*; *Sumpter et al., 2005*).

To test whether 25HC's antiviral effects in Huh7.5 cells were due to activation of ACAT, we first needed to generate an Huh7.5 cell line that lacked ACAT activity. Unlike the CHO-K1 cells used above, ACAT activity in Huh7.5 cells arises from two isoforms of ACAT, ACAT1 and ACAT2 (*Pramfalk et al., 2020*). Therefore, we used CRISPR-Cas9 technology to disrupt the genes encoding both ACAT1 (*SOAT1*) and ACAT2 (*SOAT2*) in Huh7.5 cells, and the resulting cells were designated as Huh7.5^ΔACAT cells (*Figure 7—figure supplement 1A* and *Materials and methods*). ACAT enzymatic activity, as judged by cholesteryl [¹⁴C]oleate formation, was stimulated by 25HC to an expected degree in Huh7.5 cells but not in Huh7.5^ΔACAT cells (*Figure 7—figure supplement 1B*). Consistent with the diminished ACAT activity, 25HC was unable to deplete accessible cholesterol from the PMs of the Huh7.5^ΔACAT cells (*Figure 7—figure supplement 1C*).

We first compared the infection levels of ZIKV and hCoV-OC43 in Huh7.5 and Huh7.5^ΔACAT cells after 24 hr. The ability of ZIKV to infect Huh7.5^ΔACAT cells was increased by 60% compared to infection of Huh7.5 cells (*Figure 7A*, *middle* and *right panels*, *gray bars*). In contrast, hCoV-OC43 infection of Huh7.5^ΔACAT cells was reduced by 45% compared to infection of Huh7.5 cells (*Figure 7B*, *middle* and *right panel*, *gray bars*). Since the relative infection levels vary in opposite directions, we think it is

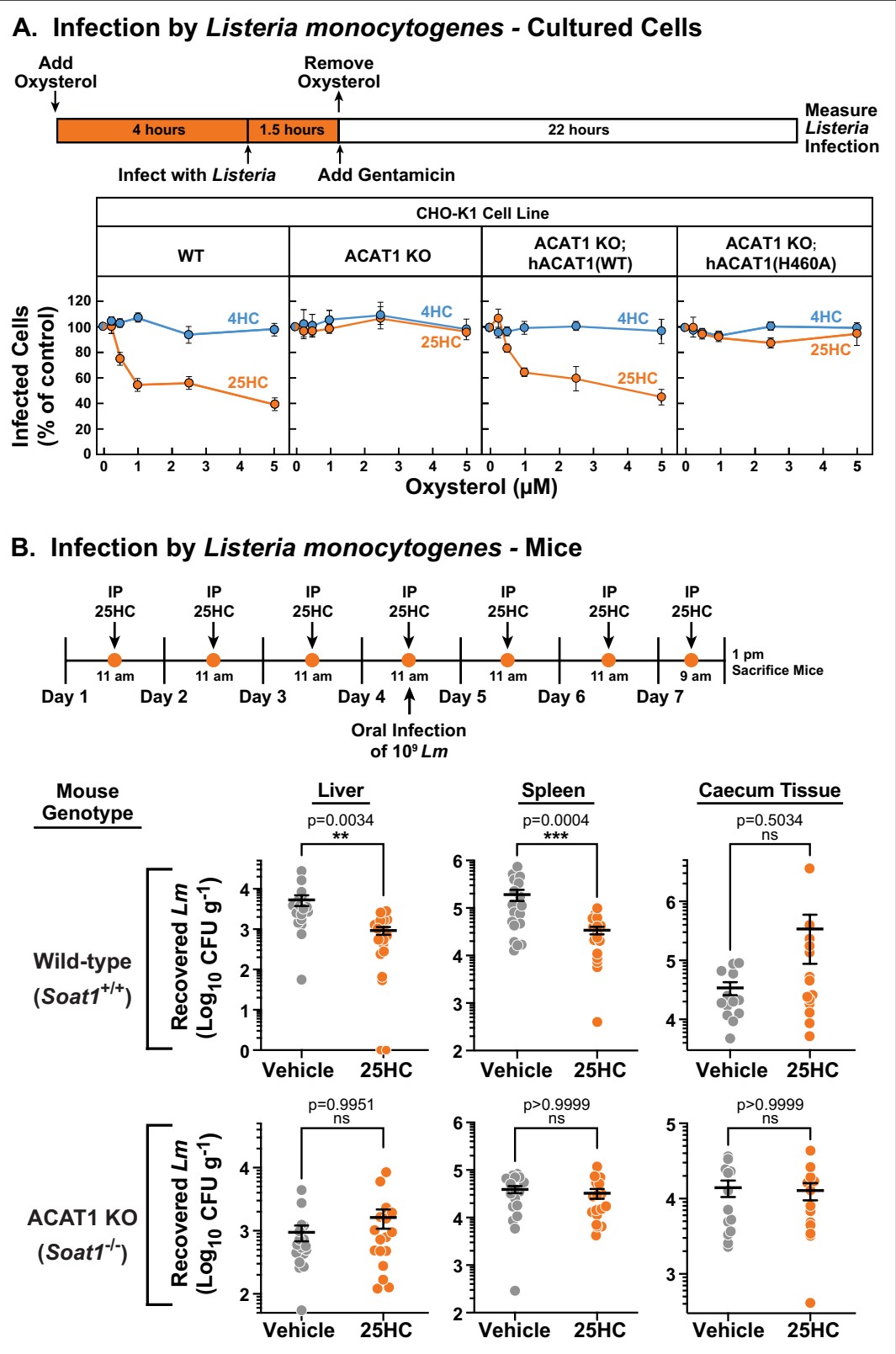

**Figure 6.** 25HC fails to protect ACAT-deficient cell lines and mice from infection by *Listeria monocytogenes*. (**A**) *Listeria* infection of CHO-K1 cells. On day 0, the indicated versions of CHO-K1 cells were set up in medium I at a density of $1\times10^5$ cells per well of a 24-well plate. On day 1, media was supplemented with either 25HC or 4HC to obtain the final concentration indicated. After incubation at 37 °C for 4 hr, cells were infected with *Listeria*

*Figure 6 continued on next page*

*Figure 6 continued*

*monocytogenes* (MOI = 1) for 90 min. Following this step, cells were washed twice with 1 ml of PBS followed by addition of 1 ml of medium I supplemented with 50 µg/ml of gentamicin to kill extracellular bacteria. After 22 hr, cells were harvested and infection levels were determined as described in *Materials and methods*. For each cell line, the infection level measured in the absence of oxysterols was set to 100% for each replicate, and all other values were normalized to this set-point. (**B**) *Listeria* infection of mice. On days 1–7, wild-type and ACAT1 KO mice were injected once daily intraperitoneally with either 25HC or ethanol. On day 4, the mice were orally infected with 1x10⁹ *Listeria monocytogenes* strain EGD harboring a mutation in Internalin A (*Lm*-InlA^m) as described in the *Materials and methods*. On day 7, 4 hr after the 25HC injection, the spleen, liver, and caecum tissues of each mouse was collected and infection levels were determined as described in *Materials and methods*. Asterisks denote levels of statistical significance (one-way analysis of variance (ANOVA) with Dunnett's correction): non-significant (ns) p>0.05; * p≤0.05; ** p≤0.01; and *** p≤0.001. *IP*, intraperitoneal injection.

The online version of this article includes the following source data and figure supplement(s) for figure 6:

**Source data 1.** Data supporting *Figure 6* and *Figure 6—figure supplement 1*.

**Figure supplement 1.** Susceptibility of wild-type and ACAT-deficient CHO-K1 cells to infection by *Listeria monocytogenes*.

unlikely that the differences are related to accessible PM cholesterol levels in Huh7.5 and Huh7.5^ΔΔACAT cells. We next tested the ability of 25HC to protect Huh7.5 cells from these viruses. 25HC treatment for 4 hr reduced ZIKV and hCoV-OC43 infection by 48% and 57%, respectively (*Figure 7A and B*, *middle panels*, *orange bars*). Strikingly, the protective effect of 25HC toward both viruses was severely attenuated in the Huh7.5^ΔΔACAT cells (*Figure 7A and B*, *right panels*, *orange bars*). In all cases, treatment with 4HC did not alter infection (*Figure 7A and B*, *blue bars*). We also noted that accessible cholesterol was depleted from the PMs of Huh7.5 cells, but not Huh7.5^ΔΔACAT cells, by 25HC treatment for 4 hr, and that 4HC had no effects (*Figure 7—figure supplement 1C*).

In summary, these studies demonstrate that depletion of PM accessible cholesterol through ACAT activation is a common mechanism underlying the anti-toxin, anti-bacterial, and anti-viral properties of 25HC.

## Discussion

Almost 50 years ago, 25HC was identified as a potent suppressor of cholesterol synthesis (*Kandutsch and Chen, 1974*; *Brown and Goldstein, 1974*). Since then, 25HC has been found to affect virtually every aspect of cholesterol homeostasis (*Brown et al., 2021*). Yet, the biological role of this oxysterol has remained mysterious since mice lacking CH25H, the enzyme that produces 25HC, have no defects in cholesterol metabolism (*Russell, 2003*). A role for 25HC in the immune system was first recognized with the finding that stimulation of macrophage Toll-like receptors (TLRs) induced expression of CH25H and production of 25HC, suppressing the production of immunoglobulin A by B cells as part of the adaptive immune response (*Bauman et al., 2009*). There is also growing evidence that 25HC exhibits potent antibacterial and antiviral properties (*Cyster et al., 2014*; *Spann and Glass, 2013*; *Blanc et al., 2013*; *Gold et al., 2014*; *Zhao et al., 2020*; *Griffiths and Wang, 2021*), but the underlying mechanisms remain unresolved. In this study, we provide a potentially unifying model for the broad immunological activities of 25HC. The model builds on recent observations that bacteria and viruses exploit a small pool of cholesterol in the host cell's PM, termed accessible cholesterol, to promote infection (*Abrams et al., 2020a*; *Zhou et al., 2020*; *Wang et al., 2020*). We show that 25HC manipulates pathways normally used to maintain cholesterol homeostasis to (i) rapidly deplete accessible PM cholesterol; and (ii) maintain this depleted state over an extended period of time to deliver long-lasting protection against bacterial and viral infection.

A striking feature of 25HC is its ability to act within 30–60 min to deplete accessible PM cholesterol (*Figure 1A*). We found that 25HC swiftly depletes accessible PM cholesterol primarily through stimulation of ACAT's enzymatic activity, and not through its effects on LXR and SREBP transcriptional pathways (*Figures 1 and 2*). Since ACAT resides in the ER, we were surprised that its activation could rapidly deplete accessible cholesterol from the membrane of a different organelle, the PM. Our studies suggest that such depletion occurs by exploiting the rapid transport pathways that normally move cholesterol between the PM and ER to maintain an equivalence of accessible cholesterol between

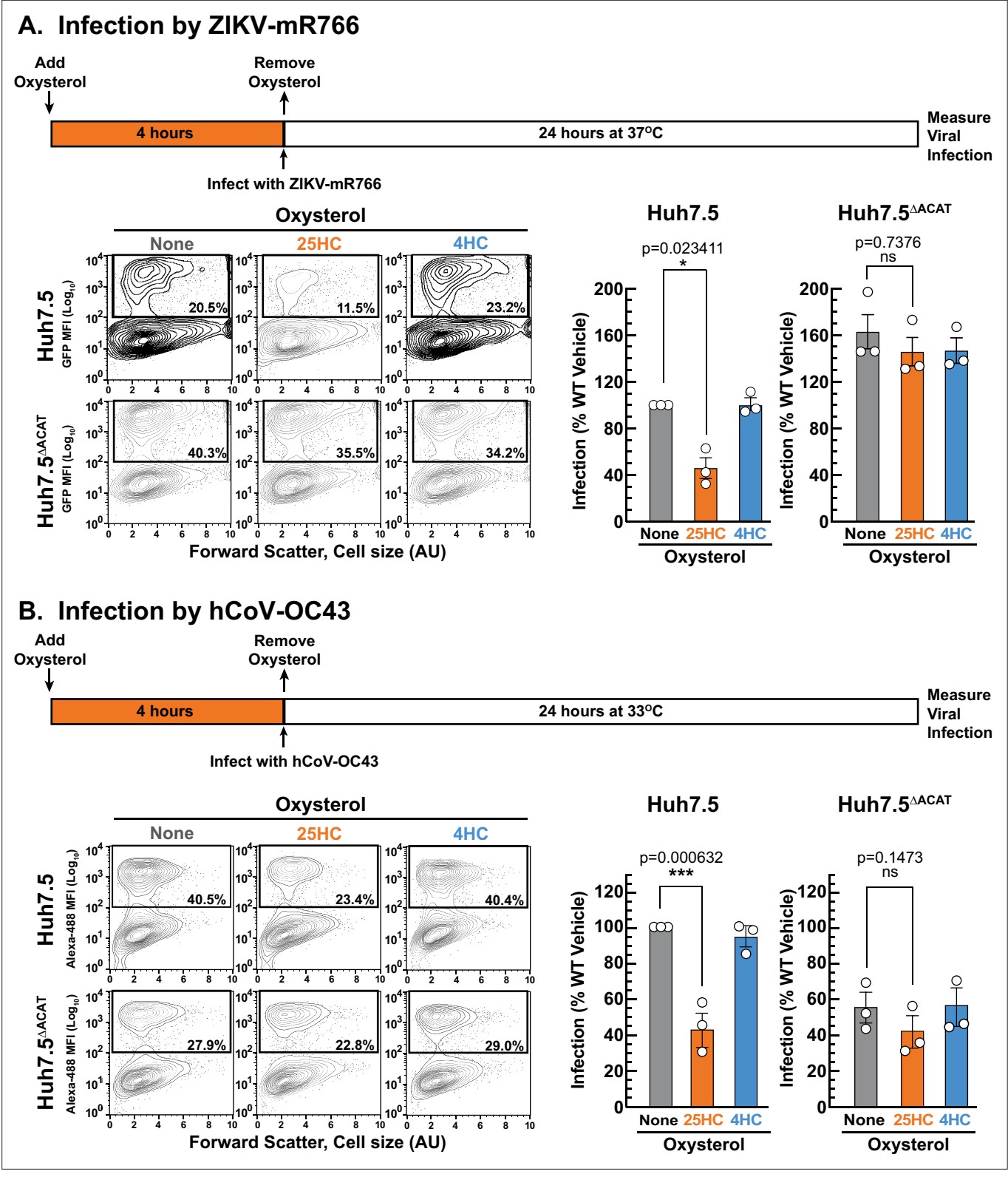

**Figure 7.** 25HC fails to protect ACAT-deficient cells from viral infection. (**A, B**) Viral infection of Huh7.5 cells. On day 0, the indicated versions of Huh7.5 cells were set up in medium E at a density of 7x10⁴ cells per well of a 24-well plate. On day 1, media was removed and replaced with 1 ml of medium E supplemented with 5 µM of the indicated oxysterol. After incubation at 37 °C for 4 hr, media was removed, and cells were infected with either hCoV-OC43 (**A**) or ZIKV-mR766 (**B**) at the indicated temperatures as described in *Materials and methods*. After 24 hr, cells were harvested, and infection levels were determined as described in *Materials and methods*. Representative flow cytometry plots of infected cells and quantification of infection

*Figure 7 continued on next page*

*Figure 7 continued*

levels (*rectangular boxes*) are shown for hCoV-OC43 (**A**) and ZIKV-mR766 (**B**). The infection value obtained for Huh7.5 cells in the absence of oxysterol treatment for each experiment was set to 100% and all other values are normalized to this set-point. Asterisks denote levels of statistical significance (one-way analysis of variance (ANOVA) with Dunnett's correction) of the unnormalized data: non-significant (ns) p>0.05; * p≤0.05; ** p≤0.01; and *** p≤0.001. In all panels, no significant differences (p>0.05) were detected in cells treated with 4HC.

The online version of this article includes the following source data and figure supplement(s) for figure 7:

**Source data 1.** Data supporting *Figure 7* and *Figure 7—figure supplement 1B*.

**Figure supplement 1.** Characterization of a Huh7.5 cell line deficient in ACAT1 and ACAT2.

**Figure supplement 1—source data 1.** Western blots corresponding to *Figure 7—figure supplement 1C*.

the two membranes (*Figure 3*). Activation of ACAT siphons off some of the ER accessible cholesterol to form cholesteryl esters, which results in an imbalance in accessible cholesterol levels between the PM and the ER. The rapid movement of cholesterol between the two membranes swiftly rectifies this imbalance and leads to a net lower level of accessible cholesterol in both membranes (*Figure 3*). Thus, 25HC exploits existing cholesterol homeostatic features to rapidly deplete accessible PM cholesterol. In line with this model, a prescient earlier study showed that 25HC activation of ACAT in macrophages reduces a pool of PM cholesterol that is accessible for modification by the enzyme cholesterol oxidase (*Tabas et al., 1988*). It remains to be determined whether 25HC activation of ACAT depletes accessible cholesterol from other cholesterol-rich organelles such as lysosomes.

A potentially confounding observation is that stimulation of mouse macrophage TLRs by lipopolysaccharides (LPS) increases CH25H expression only for a short period of time (~4 hr), after which CH25H expression declines down to baseline levels (*Bauman et al., 2009*; *Dennis et al., 2010*). How then could short-lived production of 25HC protect cells from bacterial and viral infections that occur over much longer time periods? Remarkably, we found that in cells exposed to 25HC for short periods of time (4 hr), accessible PM cholesterol levels remained low even after 22 hr (*Figure 4A*). This result was surprising since one would expect that depletion of accessible PM cholesterol would also deplete accessible cholesterol in the ER, leading to upregulation of SREBP transcription factors that would increase cholesterol synthesis and uptake to replenish accessible PM cholesterol. Indeed, such replenishment is observed when accessible PM cholesterol is depleted by treatment with a cholesterol-extracting reagent such as HPCD (*Figure 4—figure supplement 1*). In contrast, after cells were exposed to 25HC for 4 hr, the residual amounts of 25HC that remained in the cells was sufficient to maintain SREBPs in their inactive state over an extended period of 22 hr (*Figure 4A*). Thus, 25HC achieves both rapid and long-lasting depletion of accessible PM cholesterol through its concerted effects on ACAT and SREBPs (*Figure 4*). Although the panel of oxysterols studied here (*Figure 1C*) is by no means an exhaustive list, it is worth noting that the four other oxysterols that deplete accessible cholesterol from PMs through stimulation of ACAT activity (20(*R*)HC, 24(*R*)HC, 24(*S*)HC, and 27HC) also suppress the activation of SREBP transcription factors (*Figure 1H*). It is attractive to speculate that these other oxysterols may also regulate accessible PM cholesterol in other biological contexts.

The model we propose for 25HC's anti-bacterial and anti-viral functions is dependent on ACAT activation as the initiator of rapid depletion of accessible PM cholesterol. In the absence of ACAT activity, 25HC loses the ability to protect cells from lysis by cytolysins secreted by *Clostridium perfringens* and *Bacillus anthracis* (*Figure 5*), infection by *Listeria monocytogenes* (*Figure 6A*), and infection by Zika virus and coronaviruses (*Figure 7*). In mammalian cells, ACAT activity arises from two isoforms – ACAT1 and ACAT2 (*Chang et al., 2006*). While ACAT activity can be completely abolished in cultured cells through CRISPR-mediated deletion of either or both isoforms of ACAT, an animal knockout of both isoforms of ACAT is not currently available. However, 25HC's ability to protect against infection by *Listeria monocytogenes* was completely lost in ACAT1-deficient mice (*Figure 6B*). Further studies are needed to identify the cell types targeted by 25HC since the ACAT1-deficient mice lost ACAT activity in peritoneal macrophages and adrenal tissue, but not in the liver (*Meiner et al., 1996*).

An outstanding question that has yet to be answered is how pathogens exploit accessible PM cholesterol for infection and how 25HC-mediated loss of accessible PM cholesterol provides immune protection. In the case of cytolysins such as Perfringolysin O and Anthrolysin O that bind accessible cholesterol on the outer PM leaflet, oligomerize, and form lytic pores, it is plausible that depletion of accessible PM cholesterol would confer protection by preventing cytolysin binding. However, how

depletion of accessible PM cholesterol protects against intracellular pathogens such as *Listeria monocytogenes* or viruses remains largely unknown. We suspect that accessible PM cholesterol regulates the activities of one or more protein receptors involved in these processes, as has been shown for proteins in the Hedgehog signaling pathway (*Radhakrishnan et al., 2020*). It is worth noting that cholesterol has been implicated in the regulation of several immune receptor signaling complexes (*Zimmerman et al., 2016*; *Chen et al., 2022*). Thus, the principles and tools described in the current study promise to aid in uncovering the mechanisms by which accessible PM cholesterol is exploited by pathogens and is mobilized by the host immune system through 25HC and other functionally related oxysterols.

## Materials and methods

### Mice

All animal experiments were performed with the approval of the Institutional Animal Care & Use Committee (IACUC) at the University of Texas Southwestern Medical Center (UTSW; Approval Reference Number: APN102346). Mice were housed in 12 hr light/12 hr dark cycles and given ad libitum access to food and water at the UTSW Animal Resource Center. ACAT1 global knockout (ACAT1 KO) mice were obtained from the laboratory of T.Y. Chang at Dartmouth College. To produce experimental mice, we first crossed male ACAT1 KO (*Soat1$^{-/-}$*) mice with WT C57BL/6 J (*Soat1$^{+/+}$*) mice to generate heterozygous *Soat1$^{+/-}$* mice. The *Soat1$^{+/-}$* mice were then inbred crossed to generate homozygous *Soat1$^{-/-}$* and *Soat1$^{+/+}$* mice. *Soat1$^{+/+}$* littermates were inbred crossed to generate experimental WT controls, and *Soat1$^{-/-}$* mice were inbred crossed to generate experimental ACAT1 KO mice. For each experiment, 9- to 14-week-old mice (both sexes) were randomly assigned to each experimental group and the order of mouse dosing and sample collection were randomized between each group.

### Cell lines

All stock cultures of cells were grown in monolayer at 37 °C in the indicated media (see **Reagents – Culture Media** in **Method Details**). CHO-K1 cells (*Cricetulus griseus*; female; ovary) were maintained in medium B in 8.8% $CO_2$. Scap-deficient SRD13A cells (*Cricetulus griseus*; female; ovary) (*Rawson et al., 1999*) and Site-2 protease-deficient cells (*Cricetulus griseus*; female; ovary), in which expression of the nuclear forms of each of SREBP-1a, –1 c, or –2 can be induced by muristerone A (*Pai et al., 1998*), were maintained in medium B supplemented with 1 mM sodium mevalonate, 20 µM sodium oleate, and 5 µg/ml cholesterol in 8.8% $CO_2$. HEK293A cells (human; female; kidney epithelial), LXRα/β-deficient cells (human; female; kidney epithelial) (*Abrams et al., 2020a*), HEK293T (human; female; kidney epithelial), and Huh7.5 cells (human; male; liver epithelial) were maintained in medium D in 5% $CO_2$. To guard against potential genomic instability, all cell lines were passaged for less than 6 weeks, after which a fresh aliquot of cells was thawed and propagated. All cell lines were confirmed to be free of mycoplasma contamination using the LookOut Mycoplasma PCR Detection Kit (Sigma).

### Bacterial strains

For all *Listeria* infections, *Listeria monocytogenes* 10403s was grown overnight in BHI medium supplemented with 100 µg/ml streptomycin at 30 °C without shaking. *Listeria monocytogenes* EGD was grown overnight in BHI medium at 30 °C without shaking.

### Viruses

ZIKV-MR766-Venus was propagated in Huh7.5 cells as previously described (*Schwarz et al., 2016*). hCoV-OC43 was obtained from ATCC (Cat#VR-1558) and stored at –80 °C until use.

### Reagents

#### Buffers

Buffer A contains 50 mM Tris-HCl (pH 7.5), 150 mM NaCl, and 1 mM tris (2-carboxyethyl) phosphine (TCEP). Buffer B contains 10 mM Tris-HCl (pH 6.8), 100 mM NaCl, 1% (w/v) SDS, 1 mM EDTA, 1 mM EGTA, 20 µg/ml phenylmethylsulfonyl fluoride, and protease inhibitors (1 protease inhibitor cocktail tablet (Roche)/ 20 ml). Buffer C is PBS supplemented with 4% (v/v) paraformaldehyde. Buffer D is PBS supplemented with 2% (w/v) bovine serum albumin. Buffer E is PBS supplemented with 3% (w/v)

FCS. Buffer F is PBS supplemented with 1 mM EDTA. Buffer G is Hank's Balanced Salt Solution (HBSS) supplemented with 10% (v/v) FCS.

## Culture media

Medium A is a 1:1 mixture of Ham's F-12 and Dulbecco's modified Eagle's medium (DMEM) supplemented with 100 units/ml penicillin and 100 µg/mL streptomycin sulfate. Medium B is medium A supplemented with 5% (v/v) FCS. Medium C is medium A supplemented with 5% (v/v) LPDS, 50 µM sodium compactin, and 50 µM sodium mevalonate. Medium D is DMEM (high glucose) supplemented with 5% (v/v) FCS, 100 units/ml penicillin, and 100 µg/mL of streptomycin sulfate. Medium E is DMEM (high glucose) supplemented with 10% (v/v) FCS and 1 X MEM non-essential amino acids (NEAA). Medium F is medium E supplemented with 100 units/ml penicillin and 100 µg/mL streptomycin sulfate. Medium G is DMEM (high glucose) supplemented with 3% (v/v) FCS and 1 X NEAA. Medium H is medium A supplemented with 1 X NEAA, 20 mM Hepes (pH 7.4), and 4 µg/ml hexadimethrine bromide. Medium I is medium B without penicillin or streptomycin sulfate.

## ACAT-deficient cell lines

Mammalian cells express two isoforms of ACAT, ACAT1 (also designated as SOAT1) and ACAT2 (also designated as SOAT2), both of which convert cholesterol to cholesteryl esters. Previous studies have shown that ~99% of ACAT activity in CHO cells arises from ACAT1 (*Cadigan et al., 1988*; *Chang et al., 1995*), whereas ACAT activity in Huh7.5 cells arises from both ACAT1 and ACAT2 (*Pramfalk et al., 2020*). To generate cells that lacked ACAT activity, we therefore disrupted the gene encoding ACAT1 in CHO-K1 cells (*Soat1*) and the genes encoding both ACAT1 and ACAT2 in Huh7.5 cells (*SOAT1* and *SOAT2*) using CRISPR-Cas9 technology (*Ran et al., 2013*).

To generate mutant CHO-K1 cells, guide RNAs targeting exons 2 (5′-AGGAACCGGCTGTCAAAATC-3′) and 14 (5′-ATAGCTCAAGCAGACAGCGA-3′) of *Soat1* (ACAT1) (*Figure 2—figure supplement 1A*) were designed using the Benchling CRISPR guide RNA design tool (https://www.benchling.com/crispr) and cloned into the lentiviral vector lentiCRISPR v2 (Addgene 52961) (*Sanjana et al., 2014*). For lentivirus production, HEK293T cells were set up in medium E at a density of 4x10^5 cells per well of polylysine-coated six-well plates. The next day, cells were transfected with four plasmids: (i) a puromycin-selectable lentiCRISPR v2 plasmid encoding a guide targeting exon 2 of *Soat1* (0.5 µg); (ii) a blasticidin-selectable lentiCRISPR v2 plasmid encoding a guide targeting exon 14 of *Soat1* (0.5 µg); (iii) a plasmid encoding HIV gag-polymerase (0.8 µg); and (iv) a plasmid encoding vesicular stomatitis virus glycoprotein (0.2 µg). Transfections were carried out in 1 ml of medium G using X-tremeGENE 9 according to the manufacturer's protocol. After 6 hr, the media containing transfection reagents was removed and replaced with 1.5 ml of fresh medium G. After 48 hr, the media was removed and stored, and 1.5 ml of fresh medium G was added to the cells. After an additional 24 hr, media was removed and combined with the media collected after 48 hr. The pooled media was supplemented with 20 mM Hepes-NaOH (pH 7.4) and 4 µg/ml hexadimethrine bromide, a cationic polymer that increases the efficiency of lentiviral transduction, and subjected to centrifugation at 800 x g for 5 min. The supernatants containing lentiviral particle-rich media were stored at –80 °C.

To generate ACAT1-deficient cells using this lentivirus, CHO-K1 cells were set up on day 0 in medium B at a density of 1x10^5 cells per well of a 12-well plate. On day 1, media was removed and replaced with 1 ml medium H plus 120 µl of lentiviral particle-rich medium. The plate was then subjected to centrifugation at 1000 x g for 30 min at 37 °C, following which the spin-inoculated cells were placed in a 37 °C incubator with 5% CO$_2$. After 24 hr, the media was removed, cells were washed twice with PBS, following which 1 ml of medium B was added to each well. After 72 hr, transduced cells were selected by virtue of their ability to survive after incubation in medium B supplemented with 6 µg/ml puromycin and 12 µg/ml blasticidin for 14 days. During the 14-day selection period, media was removed every 2–3 days and replaced with fresh medium B supplemented with 6 µg/ml puromycin and 12 µg/ml blasticidin. Single cells were isolated by FACS and used to establish clonal cell lines. The *Soat1* genes in these lines were analyzed by genomic sequencing using primers to amplify exon 2 (5′-CTACAAGAGCTAGTTTCAGG-3′ and 5′-CCCTGTGTGTACAGTGCCTT-3′) and exon 14 (5′-TCACTCACCTTGAAGACCCA-3′ and 5′-GGGTTCCTCTCTACACACTCA-3′). Sequencing analysis revealed one cell line that had a 4 bp deletion in both alleles of exon 14 resulting in a premature stop codon (*Figure 2—figure supplement 1A*). We did not detect any disruptions in the sequence of exon

2. This cell line was designated as CHO-K1 ACAT1 KO and used for analysis of the role of ACAT1 in maintaining accessible cholesterol on PMs.

The CHO-K1 ACAT1 KO cell line was also used as template to generate two additional cell lines expressing either WT human ACAT1 or a mutant version of human ACAT1 where H460, the catalytic histidine residue, was mutated to alanine (H460A). To achieve this goal, we used Gateway cloning methods to introduce either the WT or mutant *Soat1* gene (encoding ACAT1) into the pTRIP.CMV. IVSB.ires.TagRFP destination lentiviral vector (*Schoggins et al., 2011*). The *Soat1* genes were inserted upstream of two elements in the pTrip vector, an internal ribosomal entry site and Tag-RFP. Lentiviral particle-rich media was generated as described above and used to transduce the CHO-K1 ACAT1 KO cells using similar procedures as described above for CHO-K1 cells. After 72 hr, the transduced cells were washed once with PBS, removed from each well of 12-well plates by trypsinization, and transferred to 10 cm dishes in medium B. This procedure was repeated until a total of 2x10$^7$ cells were obtained. These cells were sorted by FACS to select RFP-positive cells from which clonal cell lines were established. Two cell lines that showed robust expression of human ACAT1 (WT or H460A versions) were selected for the studies reported here. All CHO-K1 ACAT1 KO cells were maintained under identical culture conditions as the parental CHO-K1 cells.

To generate an ACAT-deficient Huh7.5 cell line, we used CRISPR/Cas technology to first delete ACAT1 in Huh7.5 cells and then deleted ACAT2 in these ACAT1-deficient cells. Guide RNAs targeting exon 6 in human *SOAT1* (ACAT1) (5'-CACCAGGTCCAAACAACGGT-3') or exon 2 in human *SOAT2* (ACAT2) (5'-GGTCCATTGTACCAAGTCCG-3') (*Figure 7—figure supplement 1A*) were designed using CHOP-CHOP (https://chopchop.cbu.uib.no/). These guide RNAs were cloned into either a puromycin-selectable (exon 6 of *SOAT1*) or a blasticidin-selectable (exon 2 of *SOAT2*) lentiviral vector lentiCRISPR v2, and lentiviruses were generated as described above.

To generate ACAT-deficient cells using these lentiviruses, Huh7.5 cells were set up on day 0 in medium E at a density of 7x10$^4$ cells per well of a 12-well plate. On day 1, media was removed and replaced with 0.4 ml medium G plus 0.1 ml of lentivirus targeting exon 6 of *SOAT1* (ACAT1). After 6 hr, 2 mL of medium F was added to each well without removing the lentivirus. After additional incubation for 72 hr, cells from three wells of the 12-well plate were collected by trypsinization and set up in a single well of a six-well plate. After 24 hr, the media was removed, cells were washed twice with PBS, and 2 ml of medium F supplemented with 8 µg/ml puromycin was added to each well. Puromycin selection was carried out for 14 days. During this selection period, media was removed every 2–3 days and replaced with 2 ml of fresh medium F supplemented with 8 µg/ml puromycin. After selection, single cells were isolated by FACS and set up in 96-well plates in 100 µl of medium F. Unfortunately, we were unable to establish clonal cell lines from these single-cell clones. To overcome this problem, 24 hr after the isolated single cells were set up in 96-well plates, we added Huh7.5 cells (1.5x10$^3$) in 100 µl of medium F supplemented with additional FCS to reach a final concentration of 40% (v/v). After 5 days, 100 µl of medium F without FCS was added to each well to replenish media lost to evaporation. This replenishment step was repeated on the 10$^{th}$ day. After 15 days, the media was removed and replaced with 200 µl of medium F supplemented with 8 µg/ml puromycin. After an additional 7 days, only lentiviral-transduced cells survived, and these cells were collected by trypsinization and set up in 24-well plates in medium F supplemented with 8 µg/ml puromycin. After further selection for 3 days, surviving cells were expanded in medium F to obtain enough cells for genomic sequencing. The *SOAT1* genes were analyzed using primers to amplify exon 6 (5'-CAGC GTATTAACGTTGTGGTGT-3' and 5'-GCCCAATGTTGAAACAGAAAAT-3'). Sequencing analysis of *SOAT1* revealed one cell line that had a 1 bp insertion in one allele and a 63 bp deletion in the other allele, resulting in premature stop codons in both cases (*Figure 7—figure supplement 1A*). This *SOAT1* (ACAT1)-deficient cell line was transduced as described above with a lentivirus targeting exon 2 of *SOAT2* (ACAT2). Selection was carried out as described above, except that medium F supplemented with 6 µg/ml blasticidin was used in all selection steps. Candidate cell lines were analyzed by genomic sequencing using primers to amplify exon 2 of *SOAT2* (5'-CAACTTCCCCTTCTAGTAGC CC-3' and 5'-CTTTATCACCAAGCCTCACTCC-3'). Sequencing analysis of *SOAT2* revealed one cell line that had a 7 bp deletion in one allele and a 1 bp insertion in the other allele, resulting in premature stop codons in both cases (*Figure 7—figure supplement 1A*). This cell line, designated as Huh7.5$^{ΔΔACAT}$, was maintained under identical culture conditions as the parental Huh7.5 cells and used for further studies involving viral infections.

## Protein purification and labeling

Recombinant His$_6$-FLAG-ALOD4 and His$_6$-Neon-FLAG-ALOD4 (designated as fALOD4-Neon) were purified by nickel chromatography followed by gel filtration chromatography as described previously (*Johnson and Radhakrishnan, 2021*). In some cases, the lone engineered cysteine on purified His$_6$-FLAG-ALOD4 was labeled with fluorescein-5-maleimide as described previously for other maleimide dyes (*Endapally et al., 2019b*) and the fluorescent protein was designated as fALOD4-fluorescein. Recombinant OlyA-His$_6$ was purified by nickel chromatography followed by gel filtration chromatography as described previously (*Endapally et al., 2019a*). In some cases, the lone engineered cysteine on purified OlyA-His$_6$ was labeled with either Alexa Fluor 647 C$_2$-maleimide or fluorescein-5-maleimide as described previously (*Endapally et al., 2019a*), and the fluorescent proteins were designated as fOlyA-647 or fOlyA-fluorescein, respectively. Recombinant His$_6$-tagged versions of full-length (FL) ALO and PFO were purified by nickel chromatography followed by gel filtration as described previously (*Gay et al., 2015*). fALOD4-Neon and fOlyA-647, in buffer A supplemented with 20% (v/v) glycerol, were flash frozen in liquid N$_2$ and stored at –80 °C for up to 6 months. The fluorescein-labeled proteins were stored in buffer A at 4 °C and used within 2 weeks. His$_6$-FLAG-ALOD4, His$_6$-ALO(FL), His$_6$-PFO(FL), and OlyA-His$_6$ were stored in buffer A at 4 °C and used within 1 month.

## Assays

### Immunoblot analysis

After indicated treatments, cells were harvested as described previously (*Johnson et al., 2019*). The following primary antibodies were used: anti-His (1:1000 dilution) to detect His$_6$-FLAG-ALOD4 and OlyA-His$_6$, anti-β actin (1:1000 dilution) to detect actin, anti-ACAT1 (1:1000 dilution) to detect ACAT1, IgG-7D4 (10 µg/ml) to detect SREBP2, and anti-FLAG (1:1000 dilution) to detect muristerone-induced nuclear forms of SREBPs. Bound antibodies were visualized using a 1:5000 dilution of either donkey anti-mouse IgG (Jackson ImmunoResearch) or goat anti-rabbit IgG (Jackson ImmunoResearch) conjugated to horseradish peroxidase. Membranes were exposed to either Phoenix Blue X-Ray film (Phoenix Research Products) or UltraCruz Autoradiography (Santa Cruz Biotechnology) at room temperature for 1–120 s for all cases.

Immunoblot signals were quantified by densitometry analysis using ImageJ as follows. In each experiment (dose curve or time curve), the anti-actin, anti-His (for ALOD4 or OlyA binding), or anti-SREBP2 immunoblot signals for each condition were quantified. The actin signal for each condition was divided by the highest value for the actin immunoblot signal in that experiment to generate relative values for actin signal. Next, the His or SREBP2 immunoblot signal for each condition was divided by the relative actin signal for that condition to generate a normalized His or SREBP2 signal for each condition. The normalized signals at the zero-timepoint or zero-concentration was set to 100% and all other values were reported relative to this value. Further details can be found in the Figure Legends.

### Cell surface binding of ALOD4 or OlyA after treatment with oxysterols

On day 0, cells were set up as indicated in the Figure Legends. On day 1, media was removed, cells were washed twice with PBS, then treated with media containing the indicated concentration of oxysterols solubilized in ethanol. In each experiment, the amount of ethanol added to each well was held constant (maximum 0.2%). After incubation for the indicated times, media was removed and replaced with media supplemented with 3 µM of either His$_6$-Flag-ALOD4 or OlyA-His$_6$. After incubation for 30 min at 37 °C, cells were washed twice with PBS, harvested in buffer B, and equal aliquots were subjected to immunoblot analysis using anti-His and anti-actin antibodies.

### Microscopy

Cells were treated with oxysterols and fluorescent sensor proteins as indicated in Figure Legends. Oxysterols were solubilized in ethanol and the amount of ethanol added during each treatment was held constant (maximum 0.2%). After treatment, cells were washed twice with PBS and then fixed with 500 µl of buffer C. After incubation at room temperature (RT) for 15 min, cells were washed four times with PBS and then treated with 500 µl of buffer D. After incubation on an orbital shaker at RT for 30 min, buffer D was removed and replaced with 500 µl of buffer D supplemented with 1 µg/ml DAPI. After further incubation on an orbital shaker at RT for 30 min, cells were washed twice with PBS and

either imaged immediately or post-fixed with 500 µl buffer C and imaged at a later time. Images were acquired on a Nikon Eclipse *Ti* epifluorescence microscope with a 60 x objective.

## Measuring ALOD4 and OlyA binding to cells by flow cytometry

Cells were set up in 24-well plates on day 0 and subjected to treatments as indicated in the Figure Legends. On day 2, after incubation with fluorescent sensor proteins at 37 °C for 30 min, media was removed, cells were washed twice with 500 µl of PBS followed by addition of 100 µl of Accumax. After incubation at 37 °C for 5 min, detached cells were transferred to a 96-well plate where each well contained 100 µl of PBS supplemented with 2% (v/v) PFA. After incubation on ice for 20 min, cells were subjected to centrifugation at 1000 x g for 5 min at RT. The supernatant was removed and the cell pellets were resuspended in 150 µl of buffer E and subjected to FACS analysis using a Stratedigm S100 EX Flow Cytometer. Data was analyzed using FlowJo software version 10.

## Quantification of intracellular levels of 25HC

Cells were set up in 24-well plates on day 0 and subjected to treatments as indicated in the Figure Legends. Mass spectrometry analysis of the cellular content of 25HC was performed as described previously (*McDonald et al., 2012*). Briefly, lipids were extracted from cellular fractions using a modified Bligh and Dyer extraction procedure with a 1:1:1 mixture of methanol, dichloromethane, and PBS. The organic extracts were saponified to generate a pool of free oxysterols, which were further purified using solid-phase extraction. Samples were evaporated under a gentle stream of nitrogen and reconstituted in 90% methanol. 25HC levels were measured with isotope dilution mass spectrometry using a deuterated analog of 25-hydroxycholesterol (d6) added to the sample prior to extraction as a standard for quantification. Lipid extracts were resolved and detected using high-performance liquid chromatography (HPLC) coupled to a triple quadrupole mass spectrometer (MS) through an electrospray ionization interface. The content of 25HC in whole cells is expressed relative to the protein content in those same cells as determined with a BCA protein assay kit.

## Cholesterol esterification

Incorporation of [$^{14}$C] oleate into cholesteryl [$^{14}$C] oleate in cultured cells after oxysterol addition was determined using previously described methods (*Goldstein et al., 1983*).

## Measuring ALOD4 binding to red blood cells

Red blood cells (RBCs) were isolated from rabbit whole blood (Innovative Research; Novi, MI) using a previously described procedure (*Chakrabarti et al., 2017*). Briefly, rabbit whole blood was subjected to centrifugation at 200 x *g* for 10 min at 4 °C. The supernatant was removed, and the cell pellet was resuspended in an equal volume of ice-cold Buffer F and then subjected to centrifugation at 500 x g for 10 min at 4 °C. The supernatant was removed, and the cell pellet was resuspended in an equal volume of ice-cold Buffer F and then subjected to centrifugation at 1000 x g for 20 min at 4 °C. After removing the supernatant, the cell pellet was resuspended in 9 volumes of Buffer F and stored at 4 °C for up to 7 days.

For each binding assay, RBCs (500 µl) were incubated without or with 5 µM of the indicated oxysterols for 4 h, or 1% (w/v) HPCD at RT for 1 hr. As controls for hemolysis, RBCs (500 µl) were incubated with 1% (v/v) Triton-X100 for 4 hr. After incubations, aliquots of the binding assays were transferred to new tubes. One aliquot (200 µl) was transferred to a tube containing 5 µl of a stock solution of fALOD4-Neon (final concentration of 1 µM) and incubated at RT for 30 min. A portion of this mixture (150 µl) was then transferred to a 96-well plate and fALOD4-Neon binding was measured by analyzing 10,000 RBCs for each condition on a Stratedigm S1000 EX Flow Cytometer (BD Biosciences). Fluorescently labeled RBCs were analyzed using FlowJo software version 10 (Ashland, OR) and geometric mean fluorescent intensity (gMFI) of each population was calculated to quantify fALOD4-Neon binding. The other aliquot (250 µl) was subjected to centrifugation (350 x g for 15 min at 20 °C) and a portion of the supernatant (125 µl) was transferred to 96-well plates. The absorbance at 540 nm was measured using a microplate reader (FLUOstar Optima) to assess hemoglobin release from lysed red blood cells. In all treatments, hemolysis was less than 10%.

## Measuring pore formation by full-length PFO or ALO

To measure the dose dependence for pore formation by PFO and ALO, cells were set up in 48-well plates on day 0 as indicated in the Figure Legends. On day 1, media was removed, cells were washed twice with HBSS, followed by addition of HBSS containing either $His_6$-PFO(FL) or $His_6$-ALO(FL). After incubation for 15 min at 37 °C, media was removed, cells were washed once with HBSS, then treated for 10 min at RT in the dark (wrapped in aluminum foil) with 100 µl of HBSS supplemented with Ghost Dye Violet 450 (1:1000), a dye that labels primary amines and is commonly used to assess membrane integrity. The labeling reaction was quenched by addition of 500 µl of buffer G for 2 min, after which the media was removed and replaced with 100 µl of Accumax, following which detached cells were transferred to a 96-well plate where each well contained 100 µl of HBSS supplemented with 2% (v/v) PFA. After incubation for 20 min at 4 °C, cells were subjected to centrifugation at 800 x g for 5 min at 20 °C. The supernatant was removed and the cell pellets were resuspended in 150 µl of buffer E and subjected to FACS analysis using a Stratedigm S100 EX Flow Cytometer. Data was analyzed using FlowJo software version 10. Pore formation was assessed as the percentage of cells that were labeled by Ghost Dye Violet 450. The highest percentage value for each cell line treated with either $His_6$-PFO(FL) or $His_6$-ALO(FL) was set to 100, and all other values for that cell line were normalized to this set-point.

To assess the effects of oxysterols on pore formation, cells were set up in 48-well plates on day 0 as indicated in the Figure Legends. On day 1, media was removed and replaced with medium B supplemented with varying concentrations of the indicated oxysterol. After incubation for 4 hr at 37 °C, media was removed, cells were washed twice with 500 µl of HBSS, followed by addition of HBSS containing either 100 pM of $His_6$-PFO(FL) or 500 pM of $His_6$-ALO(FL). Pore formation was assessed as described above. The percentage of cells with pore formation in the absence of oxysterol treatment was set to 100% for each replicate, and all other values were normalized to this set-point.

## Inhibition of Listeria monocytogenes infection by oxysterols in cultured cells

All cell culture steps were carried out in an incubator at 37 °C and 5% $CO_2$. On day 0, the indicated CHO-K1 cell lines were set up in Medium I at a density of $1x10^5$ cells per well of a 24-well plate. Also on day 0, a glycerol stock of *Listeria monocytogenes* strain 10403s expressing green fluorescent protein (GFP; kindly provided by D. Portnoy, UC Berkeley) was used to inoculate 3 ml of brain heart infusion (BHI) medium supplemented with 100 µg/ml streptomycin. After incubation for 16–18 hr at 30 °C without shaking, 1 ml of the *Listeria* culture was subjected to centrifugation at 16,000 x g for 1 min at RT. The cell pellet was resuspended in 1 ml of PBS and the centrifugation step was repeated. The resulting cell pellet was resuspended in 1 ml of PBS and the optical density at 600 nm ($OD_{600}$) was measured. Also on day 1, media for the CHO-K1 cells was supplemented with the indicated concentrations of oxysterols. After 4 hr, the cells were infected with the *Listeria* in PBS at a multiplicity of infection (MOI) of 1 for 90 min. Cells were then washed twice with 1 ml of PBS that had been pre-warmed to 37 °C, followed by addition of 1 ml of Medium I supplemented with 50 µg/ml of gentamicin to kill extracellular bacteria and prevent new infections. After 20 hr at 37 °C, media was removed, cells were washed twice with 1 ml of PBS, and harvested with 500 µl of PBS containing 0.5% (v/v) Triton X-100. To determine the extent of infection, cell lysates were serial diluted in PBS, plated on BHI agar supplemented with 100 µg/ml streptomycin, and incubated overnight at 37 °C. The next day, colonies on each plate were counted. The value for the level of infection in the absence of oxysterols was set to 100% for each replicate, and all other values were normalized to this set-point.

## Inhibition of *L. monocytogenes* infection in a mouse model

We obtained homozygous ACAT1 global knockout mice (*Meiner et al., 1996*) from Dr. T.Y. Chang (Dartmouth College). On days 1–7, 9- to 14-week-old wild-type and ACAT1 KO mice were injected once daily intraperitoneally with either 25HC (5 mg/kg in 10% ethanol) or vehicle (10% ethanol). On day 3, a glycerol stock of *Listeria monocytogenes* strain EGD harboring a mutation in Internalin A that enhances its binding affinity for murine E-cadherin (*Lm*-InlA^m) (*Wollert et al., 2007*) was used to inoculate 30 ml of BHI medium. After incubation for 16–18 h at 30 °C without shaking, the media containing *Lm*-InlA^m was subjected to centrifugation at 4000 x g for 20 min at 4 °C. The cell pellet was resuspended in 30 ml of PBS and the centrifugation step was repeated. The resulting cell pellet was resuspended in 20 ml of PBS and the $OD_{600}$ was measured to quantify *Lm*-InlA^m. An aliquot

corresponding to $10^9$ *Lm*-InlA$^m$/mouse was subjected to centrifugation at 4000 x g for 15 min at 4 °C, and the cell pellet was resuspended in a mixture of PBS and 50 mg/ml CaCO$_3$ (final ratio of 2:3, respectively). On day 4, the mice were orally infected with $1\times10^9$ *Lm*-InlA$^m$ intragastric gavage prior to being injected intraperitoneally with 25HC. On day 7, 4 hr after 25HC injection, the spleen, liver, and caecum tissues of each mouse was collected. The spleens and livers were homogenized in 2 ml of PBS, and serial dilutions of the homogenates were plated on BHI agar plates to quantify *Lm*-InlA$^m$. Caecum tissues were removed, longitudinally dissected, and caecum content was lightly scraped, followed by three sequential washes in 2 ml of PBS. Tissues were shaken in 4 ml of RPMI-1640 supplemented with 100 µg/ml of gentamicin at RT. After 30 min, tissues were vortexed for 5 sec. After an additional 30 min of shaking at RT, 10 ml of PBS was added and tissues were subjected to centrifugation at 3,000 x g for 5 min at 4 °C. The supernatant was removed, replaced with 5 ml of PBS and the centrifugation step was repeated. The supernatant was removed and replaced with 2 ml of PBS. Following homogenization of the tissues, serial dilutions of the homogenates were plated on BHI agar plates to quantify *Lm*-InlA$^m$.

## Inhibition of viral infections by oxysterols

All cell culture steps were carried out in an incubator with 5% CO$_2$ at the indicated temperatures. On day 0, Huh7.5 or Huh7.5$^{\Delta\Delta ACAT}$ cells were set up in medium F at a density of $8\times10^4$ cells per well of a 24-well plate. On day 1, the media was removed and replaced with 1 ml of medium F supplemented with 5 µM of the indicated oxysterol. After 4 hr at 37 °C, the media was removed and replaced with 200 µl of either hCoV-OC43 virus or ZIKV-MR766-Venus virus in OptiMEM and incubated at 33 °C or 37 °C, respectively. After 1 hr, 800 µl of medium F was added to each well and maintained at either 33 °C or 37 °C as indicated above. After 24 hr, cells were harvested with Accumax, mixed with a 4% (v/v) stock solution of PFA in PBS (1% final concentration), and incubated for 10 min at RT, following which cells were analyzed by FACS either immediately or stored at 4 °C and processed at a later time.

Zika-infected cells were subjected to centrifugation at 1500 x g for 5 min at RT. The supernatants were removed, and the cell pellets were resuspended in 150 µl of buffer E prior to FACS analysis.

OC43-infected cells were subjected to centrifugation at 1500 x g for 5 min at RT. The supernatants were removed, and the cell pellets were resuspended in 50 µl of BD Cytofix/Cytoperm Solution. After incubation for 20 min at RT, 150 µl of BD Wash/Perm Solution was added to each sample, after which samples were subjected to centrifugation at 1500 x g for 5 min at RT. The supernatant was removed, and cell pellets were resuspended in 200 µl of BD Wash/Perm Solution, and subjected to centrifugation at 1500 x g for 5 min at RT. After removing the supernatant, cell pellets were resuspended in 50 µl of anti-coronavirus group antigen antibody (1:50). After incubation for 30 min at RT, 150 µl of BD Wash/Perm Solution was added, and samples were subjected to centrifugation at 1500 x g for 5 min at RT. The supernatant was removed, and cell pellets were resuspended in 200 µl of BD Wash/Perm Solution and subjected to centrifugation at 1500 x g for 5 min at RT. The supernatant was removed, and cell pellets were resuspended in 50 µl of Goat anti-mouse AlexaFluor488 (1:1000). After incubation in the dark (wrapped in aluminum foil) for 30 min at RT, 150 µl of BD Wash/Perm Solution was added and samples were subjected to centrifugation at 1500 x g for 5 min at RT. The supernatant was removed, and cells were resuspended in 200 µl of BD Wash/Perm Solution and subjected to centrifugation at 1500 x g for 5 min at RT. After this final step, the supernatant was removed, and cells were resuspended in 150 µl of buffer E and analyzed on a Stratedigm S1000 EX Flow Cytometer using the GFP channel. Infection levels were determined as the percentage of Venus-positive cells (indicating Zika infection) or AlexaFluor488-positive cells (indicating OC43 infection) from at least 7500 single cells per replicate, as assessed by FlowJo software.

## Quantification and statistical analysis

All experiments were conducted at least three times with different litters of mice and with different batches of cells set up on different days. Data are shown as the mean +/- standard error of measurement (SEM), unless otherwise stated. Statistical significance was calculated using GraphPad Prism and is indicated in the figures according to the following key: non-significant (ns) $p>0.05$; * $p\leq0.05$; ** $p\leq0.01$; and *** $p\leq0.001$.

## Acknowledgements

We thank Francisco Bautista for his assistance during the early stages of generating ACAT-deficient cells; Wesley Burford for his assistance with in vivo experiments; Danya Vazquez, Bilkish Bajaj, and Chieu Nguyen for excellent technical assistance; Lisa Beatty, Camille Harry, Ijeoma Dukes, and Alex Hatton for cell culture assistance; and members of all our laboratories for helpful discussions. KAJ is the recipient of a postdoctoral fellowship from the Hartwell Foundation. We are grateful for support from the National Institutes of Health (AI158357 to AR and NMA, HL160487 to AR and JGM, AI083359 to NMA, AI158124 to JWS, and 5T32AI007520 to DBH), the Welch Foundation (I-1704 to NMA, I-1793 to AR), and the Fondation Leducq (19CVD04 to AR).

## Additional information

### Competing interests

John W Schoggins, Arun Radhakrishnan: Reviewing editor, *eLife*. The other authors declare that no competing interests exist.

### Funding

| Funder | Grant reference number | Author |
|---|---|---|
| National Institutes of Health | AI158357 | Neal M Alto<br>Arun Radhakrishnan |
| National Institutes of Health | HL160487 | Jeffrey G McDonald<br>Arun Radhakrishnan |
| National Institutes of Health | AI083359 | Neal M Alto |
| National Institutes of Health | AI158124 | John W Schoggins |
| National Institutes of Health | 5T32AI007520 | David B Heisler |
| Welch Foundation | I-1704 | Neal M Alto |
| Welch Foundation | I-1793 | Arun Radhakrishnan |
| Fondation Leducq | 19CVD04 | Arun Radhakrishnan |
| Hartwell Foundation | Fellowship | Kristen A Johnson |

The funders had no role in study design, data collection and interpretation, or the decision to submit the work for publication.

### Author contributions

David B Heisler, Kristen A Johnson, Conceptualization, Investigation, Writing – original draft, Writing – review and editing; Duo H Ma, Lishu Zhang, Michelle Tran, Chase D Corley, Investigation; Maikke B Ohlson, Conceptualization, Investigation, Writing – review and editing; Michael E Abrams, Investigation, Writing – review and editing; Jeffrey G McDonald, Conceptualization, Supervision, Funding acquisition, Investigation, Writing – review and editing; John W Schoggins, Conceptualization, Supervision, Funding acquisition, Project administration, Writing – review and editing; Neal M Alto, Conceptualization, Supervision, Funding acquisition, Writing – original draft, Project administration, Writing – review and editing; Arun Radhakrishnan, Conceptualization, Supervision, Funding acquisition, Investigation, Writing – original draft, Project administration, Writing – review and editing

### Author ORCIDs

David B Heisler  http://orcid.org/0000-0002-6482-4215
Kristen A Johnson  http://orcid.org/0000-0003-2635-7406
Duo H Ma  http://orcid.org/0000-0002-6736-551X
John W Schoggins  http://orcid.org/0000-0002-7944-6800
Neal M Alto  http://orcid.org/0000-0002-7602-3853
Arun Radhakrishnan  http://orcid.org/0000-0002-7266-7336

### Ethics

All animal experiments were performed with the approval of the Institutional Animal Care & Use Committee (IACUC) at the University of Texas Southwestern Medical Center (Approval Reference Number: APN102346).

### Decision letter and Author response

Decision letter https://doi.org/10.7554/eLife.83534.sa1
Author response https://doi.org/10.7554/eLife.83534.sa2

---

## Additional files

### Supplementary files
• MDAR checklist

### Data availability

All reagents generated in this study are available from the authors with a completed Materials Transfer Agreement. No datasets were generated in this study that required deposition in data repositories. Original, uncropped scans of all immunoblots shown in this study and raw data (including replicates and statistics) for all graphs shown are included in the source data files attached to the respective figures.

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

# Appendix 1

## Appendix 1—key resources table

| Reagent type (species) or resource | Designation | Source or reference | Identifiers | Additional information |
|---|---|---|---|---|
| Gene (*Cricetulus griseus*) | *Soat1* | NCBI Gene | Gene ID:100689317 | |
| Gene (*H. sapiens*) | *SOAT1* | NCBI Gene | Gene ID:6646 | |
| Gene (*H. sapiens*) | *SOAT2* | NCBI Gene | Gene ID:8435 | |
| Strain, strain background (*Mus musculus*) | C57BL/6 J | The Jackson Laboratory | Cat#000664 | |
| Strain, strain background (*M. musculus*) | Wild Type C57BL/6J$^{Soat1+/+}$ | This paper | | Experimental Wild-type mice |
| Strain, strain background (*M. musculus*) | Mouse: ACAT1$^{+/-}$: C57BL/6J$^{Soat+/-}$ | This paper | | Heterozygous ACAT1 mice |
| Strain, strain background (*M. musculus*) | Mouse: ACAT1$^{-/-}$: C57BL/6J$^{Soat1-/-}$ | This paper | PMID:8943057 | Experimental ACAT1 KO mice |
| Strain, strain background (*Listeria monocytogenes*) | 10403s | PMID:3114382 | | Gift from Dr. Daniel Portnoy, University of California, Berkeley |
| Strain, strain background (*L. monocytogenes*) | EGD InlA$^{mut}$ | PMID:17540170 | | Gift from Dr. Wolf-Dieter Schubert, University of Pretoria, South Africa |
| Strain, strain background (*Escherichia coil*) | BL21 (DE3) pLysS | Invitrogen | Cat#C606003 | Chemically competent cells |
| Strain, strain background (*Coronaviridae*) | hCoV-OC43 | ATCC | Cat# VR-1558 | |
| Cell line (*C. griseus*) | CHO-K1 | ATCC | CAT#CCL-61 | |
| Cell line (*C. griseus*) | CHO-K1 ACAT1 KO | This paper | | Dr. Arun Radhakrishnan (UTSW) |
| Cell line (*C. griseus*) | CHO-K1 ACAT1 KO; hACAT1(WT) | This paper | | Dr. Arun Radhakrishnan (UTSW) |
| Cell line (*C. griseus*) | CHO-K1 ACAT1 KO; hACAT1(H460A) | This paper | | Dr. Arun Radhakrishnan (UTSW) |
| Cell line (*C. griseus*) | M19 | PMID:9748295 | | Dr. Arun Radhakrishnan (UTSW) |
| Cell line (*C. griseus*) | SRD13A | PMID:10497220 | | Dr. Arun Radhakrishnan (UTSW) |
| Cell line (*H. sapiens*) | HEK293A | PMID:32284563 | | Dr. Neal Alto (UTSW) |
| Cell line (*H. sapiens*) | HEK293A LXRα;LXRβ-deficient cells | PMID:32284563 | | Dr. Neal Alto (UTSW) |
| Cell line (*H. sapiens*) | HEK293T | PMID:21478870 | | Dr. John Schoggins (UTSW) |
| Cell line (*H. sapiens*) | Huh7.5 | PMID:21478870 | | Dr. John Schoggins (UTSW) |
| Cell line (*H. sapiens*) | Huh7.5$^{ΔACAT}$ | This paper | | Dr. Neal Alto (UTSW) |
| Transfected construct (human) | pCDNA6.ZikaMR766.Venus3115Intron HDVr | PMID:27704051 | | Dr. Matthew Evans (Icahn School of Medicine at Mount Sinai) |
| Transfected construct (human) | pGag-pol | Other | | Dr. Charles Rice (Rockefeller University) |
| Transfected construct (human) | pLentiCRISPRv2-Blast | Addgene | Cat#98293 | |
| Transfected construct (human) | pLentiCRISPRv2-Puro | Addgene | Cat#98290 | |

*Appendix 1 Continued on next page*

*Appendix 1 Continued*

| Reagent type (species) or resource | Designation | Source or reference | Identifiers | Additional information |
|---|---|---|---|---|
| Transfected construct (human) | pTRIP.CMV.IVSB.ires.TagRFP | PMID:21478870 | | Dr. John Schoggins (UTSW) |
| Transfected construct (human) | pTRIP.CMV.hACAT1.ires.TagRFP | This paper | | Dr. Arun Radhakrishnan (UTSW) |
| Transfected construct (human) | pTRIP.CMV.hACAT1(H460A).ires.TagRFP | This paper | | Dr. Arun Radhakrishnan (UTSW) |
| Transfected construct (human) | pVSV-Glycoprotein | Other | | Dr. Charles Rice (Rockefeller University) |
| Antibody | anti-mouse IgG (H+L) (Donkey Polyclonal Peroxidase-AffiniPure) | Jackson ImmunoResearch | Cat#715-035-150; RRID: AB_2340770 | WB (1:5,000) |
| Antibody | anti-rabbit IgG (H+L) (Goat Polyclonal Peroxidase AffiniPure) | Jackson ImmunoResearch | Cat#111-035-003; RRID:AB_2313567 | WB (1:5,000) |
| Antibody | anti-Coronavirus Group Antigen, nucleoprotein of OC-43, 229E strain, clone 542-7D (Mouse Monoclonal) | Millipore Sigma | Cat# MAB9013; RRID:AB_95425 | Flow cytometry (1:50) |
| Antibody | anti-FLAG M2 clone (Mouse Monoclonal) | Sigma-Aldrich | F1804; RRID:AB_262044 | WB (1:1,000) |
| Antibody | anti-His Tag, clone HIS.H8 (Mouse Monoclonal) | Millipore | Cat#05–949; RRID:AB_492660 | WB (1:1,000) |
| Antibody | anti-SREBP2 (Mouse Monoclonal) | Ref. (95) | IgG-7D4 | WB (10 µg/ml) |
| Antibody | anti-Mouse IgG (H+L) Cross-Adsorbed Secondary Antibody, Alexa Fluor 488 (Mouse Polyclonal Goat) | Thermo Fisher Scientific | Cat#A-11001; RRID:AB_2534069 | Flow cytometry (1:1,000) |
| Antibody | anti-ACAT1 (Rabbit Polyclonal) | Novus | Cat#NB400-141; RRID:AB_10001588 | WB (1:1,000) |
| Recombinant DNA reagent | His$_6$-ALO(FL) in pRSET-B | PMID:25809258 | | Dr. Arun Radhakrishnan (UTSW) |
| Recombinant DNA reagent | His$_6$-FLAG-ALOD4 in pRSET-B | PMID:33712199 | | Dr. Arun Radhakrishnan (UTSW) |
| Recombinant DNA reagent | His$_6$-Neon-FLAG-ALOD4 in pRSET-B | PMID:33712199 | | Dr. Arun Radhakrishnan (UTSW) |
| Recombinant DNA reagent | OlyA-His$_6$ in pRSET-B | PMID:33712199 | | Dr. Arun Radhakrishnan (UTSW) |
| Recombinant DNA reagent | His$_6$-PFO(FL) in pRSET-B | PMID:25809258 | | Dr. Arun Radhakrishnan (UTSW) |
| Sequence-based reagent | CHO-K1 *Soat1* gRNA for Exon 2 | This paper | Target sequence | AGGAACCGGCTGTCAAAATC |
| Sequence-based reagent | CHO-K1 *Soat1* gRNA for Exon 14 | This paper | Target sequence | ATAGCTCAAGCAGACAGCGA |
| Sequence-based reagent | Huh7.5 *SOAT1* gRNA for Exon 6 | This paper | Target sequence | CACCAGGTCCAAACAACGGT |
| Sequence-based reagent | Huh7.5 *SOAT2* gRNA for Exon 2 | This paper | Target sequence | GGTCCATTGTACCAAGTCCG |
| Sequence-based reagent | CHO-K1 *Soat1* Exon 2 Forward | This paper | PCR Primer | CTACAAGAGCTAGTTTCAGG |
| Sequence-based reagent | CHO-K1 *Soat1* Exon 2 Reverse | This paper | PCR Primer | CCCTGTGTGTACAGTGCCTT |
| Sequence-based reagent | CHO-K1 *Soat1* Exon 14 Forward | This paper | PCR Primer | TCACTCACCTTGAAGACCCA |
| Sequence-based reagent | CHO-K1 *Soat1* Exon 14 Reverse | This paper | PCR Primer | GGGTTCCTCTCTACACACTCA |

*Appendix 1 Continued on next page*

*Appendix 1 Continued*

| Reagent type (species) or resource | Designation | Source or reference | Identifiers | Additional information |
|---|---|---|---|---|
| Sequence-based reagent | Huh7.5 *Soat1* Exon 6 Forward | This paper | PCR Primer | CAGCGTATTAACGTTGTGGTGT |
| Sequence-based reagent | Huh7.5 *Soat1* Exon 6 Reverse | This paper | PCR Primer | GCCCAATGTTGAAACAGAAAAT |
| Sequence-based reagent | Huh7.5 *Soat2* Exon 2 Forward | This paper | PCR Primer | CAACTTCCCCTTCTAGTAGCCC |
| Sequence-based reagent | Huh7.5 *Soat2* Exon 2 Reverse | This paper | PCR Primer | CTTTATCACCAAGCCTCACTCC |
| Commercial assay or kit | Cytofix/Cytoperm Fixation/Permeabilization Kit | BD Biosciences | Cat#554714; RRID:AB_2869008 | |
| Commercial assay or kit | LookOut Mycoplasma PCR Detection Kit | Millipore Sigma | Cat# MP0035 | |
| Commercial assay or kit | Microplate BCA Protein Assay Kit | Thermo Fisher Scientific | Cat#23252 | |
| Chemical compound, drug | 19-hydroxycholesterol | Steraloids | Cat#C6470-000 | |
| Chemical compound, drug | 2-Hydroxypropyl-β-cyclodextrin (HPCD) | Cyclodextrin Technologies Development | Cat#THPB-P | |
| Chemical compound, drug | 20(*R*)-hydroxycholesterol | Avanti Polar Lipids | Cat#700156 | |
| Chemical compound, drug | 22(*R*)-hydroxycholesterol | Avanti Polar Lipids | Cat#700058 | |
| Chemical compound, drug | 22(*S*)-hydroxycholesterol | Avanti Polar Lipids | Cat#700057 | |
| Chemical compound, drug | 24(*R*)-hydroxycholesterol | Avanti Polar Lipids | Cat#700071 | |
| Chemical compound, drug | 24(*S*)-hydroxycholesterol | Avanti Polar Lipids | Cat#700061 | |
| Chemical compound, drug | 25-hydroxycholesterol | Avanti Polar Lipids | Cat#700019 | |
| Chemical compound, drug | 25-hydroxycholesterol-3-sulfate | Avanti Polar Lipids | Cat#700017 | |
| Chemical compound, drug | 25-hydroxycholesterol-d6 | Avanti Polar Lipids | Cat#700053 | |
| Chemical compound, drug | 27-hydroxycholesterol | Avanti Polar Lipids | Cat#700021 | |
| Chemical compound, drug | 4′,6-Diamidino-2-phenylindole dihydrochloride (DAPI) | Millipore Sigma | Cat#D8417 | Microscopy (1 µg/ml) |
| Chemical compound, drug | 4β-hydroxycholesterol | Avanti Polar Lipids | Cat#700036 | |
| Chemical compound, drug | 7α-hydroxycholesterol | Avanti Polar Lipids | Cat#700034 | |
| Chemical compound, drug | Accumax | Innovative Cell Technologies | Cat#AM105 | |
| Chemical compound, drug | Alexa Fluor 647 $C_2$-maleimide | Thermo Fisher Scientific | Cat#A20347 | |
| Chemical compound, drug | Blasticidin S HCl | Thermo Fisher Scientific | Cat#A1113903 | |
| Chemical compound, drug | Bovine serum albumin | Millipore Sigma | Cat#P0834 | |

*Appendix 1 Continued on next page*

*Appendix 1 Continued*

| Reagent type (species) or resource | Designation | Source or reference | Identifiers | Additional information |
|---|---|---|---|---|
| Chemical compound, drug | Brain Heart Infusion Agar | Thermo Fisher Scientific | Cat#DF0418 | |
| Chemical compound, drug | Brain Heart Infusion Broth | Thermo Fisher Scientific | Cat#DF0037 | |
| Chemical compound, drug | Cholesterol | Millipore Sigma | Cat#C8667 | |
| Chemical compound, drug | cOmplete, EDTA-free Protease Inhibitor Cocktail | Roche | Cat# 05056489001 | |
| Chemical compound, drug | Dulbecco's Modified Eagle Medium (DMEM) – high glucose | Sigma | Cat#D6429 | |
| Chemical compound, drug | Dulbecco's Modified Eagle Medium (DMEM) – low glucose | Sigma | Cat#D6046 | |
| Chemical compound, drug | Dulbecco's Modified Eagle Medium (DMEM)/F-12 (1:1 mixture) | Corning | Cat#10–090-CV | |
| Chemical compound, drug | Dulbecco's phosphate buffered saline (PBS) | Thermo Fisher Scientific | Cat#MT21031CV | |
| Chemical compound, drug | Fetal Calf Serum | Sigma-Aldrich | Cat#F2442 | |
| Chemical compound, drug | Fluorescein-5-maleimide | Thermo Fisher Scientific | Cat#62245 | |
| Chemical compound, drug | Gentamicin | Quality Biological | Cat#120-098-661 | |
| Chemical compound, drug | Ghost Dye Violet 450 | Tonbo biosciences | Cat#13–0863 | |
| Chemical compound, drug | Hanks Balanced Salt Solution (HBSS) | Millipore Sigma | Cat#14175–095 | |
| Chemical compound, drug | Hexadimethrine bromide | Thermo Fisher Scientific | Cat#107689 | |
| Chemical compound, drug | MEM Non-Essential Amino Acids Solution (100 X) | Thermo Fisher Scientific | Cat#11140050 | |
| Chemical compound, drug | Methyl-β-cyclodextrin, randomly methylated (MCD) | Cyclodextrin Technologies Development | Cat#TRMB-P | |
| Chemical compound, drug | Muristerone A | Sigma-Aldrich | Cat#M7888 | |
| Chemical compound, drug | Opti-MEM | Thermo Fisher Scientific | Cat#31985062 | |
| Chemical compound, drug | Paraformaldehyde | Alfa Aesar | Cat#43368 | |
| Chemical compound, drug | Penicillin/Streptomycin | Gibco | Cat#15140–122 | |
| Chemical compound, drug | Phenylmethylsulfonyl fluoride (PMSF) | Goldbio | Cat#P-470–25 | |
| Chemical compound, drug | Puromycin | Thermo Fisher Scientific | Cat#A1113803 | |
| Chemical compound, drug | Rabbit Whole Blood | Innovative Research | Cat# IGRBWBK2E10ML | |
| Chemical compound, drug | Sodium compactin | PMID:624722 | N/A | |
| Chemical compound, drug | Sodium mevalonate | PMID:624722 | N/A | |

*Appendix 1 Continued on next page*

*Appendix 1 Continued*

| Reagent type (species) or resource | Designation | Source or reference | Identifiers | Additional information |
|---|---|---|---|---|
| Chemical compound, drug | Sandoz 58–035 (SZ58-035) | Millipore Sigma | Cat#S9318 | |
| Chemical compound, drug | Tris (2-carboxyethyl) phosphine Hydrochloride (TCEP) | Goldbio | Cat#TCEP1 | |
| Software, algorithm | Benchling CRISPR Guide RNA design tool | Benchling | https://www.benchling.com/crispr | |
| Software, algorithm | CHOP-CHOP | PMID:31106371 | https://chopchop.cbu.uib.no/ | |
| Software, algorithm | FlowJo | BD (Becton, Dickinson & Co.) | http://www.flowjo.com | |
| Software, algorithm | ImageJ | PMID:22930834 | https://imagej.nih.gov/ij/ | |
| Other | Eclipse Ti epifluorescence microscope | Nikon Inc. | N/A | Instrument used for microscopy imaging studies. |
| Other | S1000 Flow Cytometer | Stratedigm Inc. | NA | Instrument used for flow cytometry analysis. |

