## [Editor Report]

This paper provides important and fundamental insights into the mechanisms by which 25-hydroxycholesterol, which is known to be rapidly produced in macrophages and other cells during acute infections, acts to protect cells and animals from infectious processes. The authors provide compelling evidence that the cholesterol esterifying enzyme acylCoA:cholesterol acyltransferase (ACAT) that is induced by 25-hydroxycholesterol promotes depletion of an accessible pool of plasma membrane cholesterol, producing anti-microbial effects. The work will be of interest to those studying infection and cholesterol regulation of cellular processes.

---

## [Decision Letter]

**Decision letter after peer review:**

Thank you for submitting your article "A concerted mechanism involving ACAT and SREBPs by which oxysterols deplete accessible cholesterol to restrict microbial infection" for consideration by *eLife*. Your article has been reviewed by 3 peer reviewers, and the evaluation has been overseen by a Reviewing Editor and Suzanne Pfeffer as the Senior Editor. The following individuals involved in the review of your submission have agreed to reveal their identity: William Griffiths (Reviewer #1); Alan R. Tall (Reviewer #3).

You will be pleased to see below that the reviewers were generally quite positive. Nevertheless, they felt that the following issues need to be addressed in a revised manuscript:

1. Continued activation of ACAT could play a role in maintaining low cholesterol in the cells. This could be tested by using an ACAT inhibitor at later times in the chase after the initial 4-hour exposure to 25HC in the medium.

2. The authors used a high concentration of 25HC for their time course studies. If possible, it would be useful to know how the cellular concentration of 25HC compares with the amount generated in response to an infection. Is the 5 μM concentration relevant?

3. Please test whether ACAT inhibitors reduce this effect on cholesterol levels in macrophages, which are the relevant cell type.

4. Please quantify the western blots.

5. To provide some context for the concordant changes in ER and plasma membrane cholesterol, it would be helpful to mention that several studies have documented rapid equilibration of cholesterol among membranes.

*Reviewer #1 (Recommendations for the authors):*

1. I would recommend that the abbreviation SOAT is also included. ACAT can be confused with acetyl-coenzyme A acetyltransferase.

2. Page 5 & 6 and Figure 1, it would be much easier if the authors used simple abbreviations, 4βHC, 7αHC, 19HC rather than A, B, C, etc.

3. Figure 1. 20(R)-Hydroxycholesterol is an easier chemical nomenclature to follow than 20α-hydroxycholesterol. For completeness, I would define the stereochemistry at C25 in 27-hydroxycholesterol.

4. In the caption of Figure 4 it might be worth adding a note that a total of 25-HC was measured as a surrogate for free 25HC which I assume is the bioactive molecule. Or perhaps my assumption is wrong.

5. One thing I always wonder about is the oxysterol content of the medium, I guess it is low compared to the added 25HC, but I am guessing and don't really know.

Overall, a really nice paper that was a pleasure to read!

*Reviewer #2 (Recommendations for the authors):*

1. The paper only discusses cholesterol in the plasma membrane and the ER. However, other compartments, including the Golgi and the endocytic recycling compartment (ERC) also contain cholesterol. In CHO cells the ERC contains over 30% of the cellular cholesterol (DOI 10.1074/jbc.M1088612). There are also papers describing rapid (t_1/2_ 15 min) transport among these organelles, which is probably relevant for understanding the results in this paper. For example, cyclodextrin reduction of cholesterol will be compensated by cholesterol taken from several organelles – not just the ER. The rapid re-equilibration of cholesterol among membranes is not surprising given the rapid transport mediated by various sterol transporters.

2. Several of the figures show gels, and comparisons are made among conditions. It would be very helpful to have some quantification of these results and some indication of reproducibility.

3. For Figure 4, they used 5 μM 25HC, which is a high dose and probably is responsible for the long-lasting effects. At 8 hours, there is still about half the initial dose of 25HC in the cells, and they showed in Figure 1 that this would have a significant effect. It is likely that this amount of 25HC will still stimulate ACAT, preventing the cells from recovering. This hypothesis is easily tested with an ACAT inhibitor administered during the chase period. Thus, it is not just the continued effect on SREBP that is important.

4. They mention the effects on macrophages at the bottom of page 13. This could be tested easily with ACAT inhibitors in macrophages.

5. In Figure 7, it is confusing to change the scale for each panel.

*Reviewer #3 (Recommendations for the authors):*

A point for discussion or future investigation could be that the lingering effect of 25-OH cholesterol is not completely defined as to whether this reflects an ongoing effect of 25-OH cholesterol to bind Insigs and inhibit cholesterol biosynthesis or an effect on ACAT. In Figure 4D did ACAT1 KO prevent the effect of 25-OH cholesterol on the suppression of nuclear SREBP2? Would there be a way to more directly implicate sustained binding of low concentrations of 25-OH cholesterol to Insigs? It would also be helpful to know if concentrations of 25-OH cholesterol used in experiments or observed in cells are relevant to those likely to be achieved in the response to infectious agents in vivo.

Although the authors provide a focused and convincing discussion on the role of PM cholesterol in infectious processes, there are broader aspects of 25-OH cholesterol biology that are relevant to infectious processes that could be discussed and the topic broadened. In particular, Reboldi et al. (Science 2014) showed that 25-OH cholesterol acts to suppress the priming and activity of inflammasomes. This results in reduced inflammatory death but increased susceptibility to infection with *L. monocytogenes*. Although not necessarily contradictory, it would be interesting for the authors to discuss these earlier results in the context of their own findings.

Specific Points:

Line 96. Please cite the original report (Costet P et al., JBC 2000).

Lines 227-231. In Figure 2A ALOD4 binding seems somewhat reduced by 25-OH cholesterol in ACAT KO cells, especially after correction for actin. The same seems true when inactive ACAT is transected in the cells (bottom rows). How do the authors explain this? In Figure 2B the fOlyA fluorescence seems quite variable.

---

## [Author Response]

The reviewers have discussed their reviews with one another, and the Reviewing Editor has drafted this to help you prepare a revised submission.You will be pleased to see below that the reviewers were generally quite positive. Nevertheless, they felt that the following issues need to be addressed in a revised manuscript (together with the minor suggestions found in their detailed reviews):1. Continued activation of ACAT could play a role in maintaining low cholesterol in the cells. This could be tested by using an ACAT inhibitor at later times in the chase after the initial 4-hour exposure to 25HC in the medium.

We thank the reviewers for suggesting this elegant experiment, the results of which are shown in a new Figure 4C. We treated CHO-K1 cells with 5 µM of 25HC for 4 h, which completely abolished accessible cholesterol from the PMs (compare lane 2 to lane 1). We then removed the 25HC-containing media, washed the cells, and then added back media without 25HC. Similar to what we noted previously, no replenishment of accessible PM cholesterol was detected even after a 22 h recovery period (top two panels of Figure 4C). However, when we included 10 µM of an ACAT inhibitor (SZ58-035) during this recovery period, we began to detect accessible PM cholesterol after 8 h and ~60% of the initial level of accessible cholesterol was restored after 22 h (bottom two panels of Figure 4C and quantification in Figure 4 —figure supplement 2B). This result confirms the reviewers’ prediction that continued activation of ACAT by the lingering intracellular 25HC also plays a role in depleting accessible PM cholesterol over the longer time periods.

We have now carefully combed through our manuscript and modified all instances where we ascribed the long-term effects solely to 25HC’s effects on suppressing SREBP activation to also reflect ACAT’s contribution to the long-term depletion effects (tracked changes in revised manuscript). We are grateful to the reviewers for raising this point as it has helped clarify our mechanistic interpretations.

2. The authors used a high concentration of 25HC for their time course studies. If possible, it would be useful to know how the cellular concentration of 25HC compares with the amount generated in response to an infection. Is the 5 μM concentration relevant?

We would also love to know the amount of 25HC generated by a cell during infection, but as the reviewers can appreciate, measuring the concentration of 25HC in a particular cell is technically challenging. Even so, we can make some estimates based on measurements reported in the literature by us and others.

Two different studies (PMID: 32284563 and PMID: 23273843) reported that upon interferon-γ stimulation (25-500 U/ml for 12-24 h), the amount of 25HC secreted by bone marrow-derived macrophages (BMDMs) into the extracellular media accumulated to a final concentration of ~0.4 µM. It is plausible that the concentration of 25HC inside of the 25HC-producing cell is much higher than the 0.4 µM value measured upon dilution into the extracellular medium.

One of the previous studies (PMID: 23273843) did also measure the intracellular 25HC produced after interferon-γ stimulation and reported a value of 0.2 nmol per 10^6^ cells. Another group conducted a morphological analysis of macrophages and reported that their volume was 1000-5000 µm^3^ (PMID: 9400735). Using these estimates, the intracellular concentration of 25HC upon interferon-γ stimulation could range from 40-200 µM.

The concentration of 5 µM that we use in many of our studies is between these bookend values and thus may not be completely unreasonable. We look forward to future technical breakthroughs which will allow a more precise answer to this question.

3. Please test whether ACAT inhibitors reduce this effect on cholesterol levels in macrophages, which are the relevant cell type.

In this paper, we have used CRISPR-Cas9 knockout models to show that 25HCmediated depletion of accessible PM cholesterol is dependent on ACAT activity in hamster cells (CHO-K1) and a human hepatocyte cell line (Huh7.5). To answer the reviewer question about whether a similar ACAT-dependence operated in macrophage cells, we used a commercially available ACAT inhibitor (SZ58-035). We first tested whether SZ58-035 would inhibit ACAT activity in RAW macrophage cells (mouse). Using our standard ACAT activity assay (described in Figure 1F), we found that 25HCstimulated ACAT activity was partially reduced from 4.2 nmol/mg/h to 1.4 nmol/mg/h when the cells were pre-treated with 10 µM SZ58-035 (see Author response image 1 – left 4 columns). Treatment with higher concentrations of SZ58-035 resulted in cell death. In contrast to the results with RAW cells, we observed almost complete inhibition of ACAT activity by 10 µM SZ58-035 in CHO-K1 cells, from 6.0 to 0.1 nmol/mg/h (Author response image 1 – right 4 columns).

Even though SZ58-035 could not completely inhibit ACAT activity, we tested whether its partial inhibition of ACAT would affect 25HC’s ability to deplete accessible cholesterol from RAW cell PMs. When the macrophage cells were treated with 25HC in the absence of SZ58-035, we observed a dose-dependent reduction in accessible cholesterol levels (similar to other cell types) (Author response image 1 – top 2 panels). In the presence of the ACAT inhibitor, we observed that 25HC’s effects were greatly reduced, suggesting that even partial inhibition of ACAT activity was sufficient to inhibit 25HC’s effects in macrophage cells (Author response image 1 – bottom 2 panels). While this result is suggestive, a rigorous answer to the reviewer’s question will require generation of a macrophage cell line where both ACAT1 and ACAT2 have been eliminated by CRISPRCas9 methods or even better would be a comparison of bone marrow-derived macrophages (BMDMs) from ACAT1^+/+^/ACAT2^+/+^ and ACAT1^-/-^/ACAT2^-/-^ mice. We are currently working on generating these models and will report the results in a future study.

**Author response image 1. sa2fig1:** Inhibition of ACAT activity and depletion of accessible PM cholesterol by 25HC in RAW macrophage cells. (A) ACAT activity [same procedure as in Figure 1F]. On day 0, cells were set up either in medium D (RAW) or medium B (CHO-K1) at a density of 2.5 x 10^5^ cells per 60-mm dish. One day 2, media was removed, cells were washed twice with PBS followed by addition of 2 ml of media containing lipoprotein-deficient serum and compactin, without or with 10 µm ACAT inhibitor SZ58-035. On day 3, media was removed, cells were washed with PBS followed by addition of 1 ml of same media supplemented with 25HC [in the absence of or presence of SZ58-035]. After incubation at 37°C for 1 h, each dish was supplemented with 0.2 mm sodium [^14^C]oleate and incubated at 37°C for an additional 2 h, after which cells were harvested and levels of [^14^C]cholesteryl oleate were measured as described in the methods. Each column represents the mean of cholesterol esterification measurements (in triplicate) from two independent experiments. (B) ALOD4 binding. On day 0, RAW macrophage cells were set up in medium D containing 10% FCS at a density of 7.5 x 10^4^ cells per well of a 24-well plate. One day 1, media was removed, cells were washed twice with PBS followed by addition of 200 µl medium D (with 10% FCS) without or with 10 µm of SZ58-035. After 18 hours, media was replaced with 200 µl of the same media supplemented with the indicated concentrations of 25HC, without or with 10 µm of SZ58-035. After 1 hour, media was removed and replaced with 200 µl media supplemented with 3 µm His_6_-FLAG-ALOD4. After incubation at 37°C for 30 min, cells were washed twice with PBS, harvested and subjected to immunoblot analysis as described in methods.

4. Please quantify the western blots.

We have now quantified all the Western blots in the main Figures of our paper and presented these results in 12 new data panels in 6 new Figure supplements (associated with main Figures 1, 2, 3, and 4). In all cases, the quantifications from 3-5 replicate experiments confirm the visual interpretations that we had made based on the representative Western blots.

5. To provide some context for the concordant changes in ER and plasma membrane cholesterol, it would be helpful to mention that several studies have documented rapid equilibration of cholesterol among membranes.

We thank the reviewer for pointing out this oversight on our part, it was not our intention to ignore the vast body of work on this topic. We had mentioned that previous studies have pointed out that rapid intracellular cholesterol transport could re-equilibrate intracellular accessible cholesterol and had referenced four out of the many underlying studies that made this point (Ref. 16, 48-50; lines 278 -282 in tracked changes version). We have now included an additional reference to an informative review on this subject (PMID: 19286471; Ref. 51). We have also included a sentence in the Discussion addressing the possibility that 25HC treatment could also deplete accessible cholesterol from other organelles (lines 547-548 in tracked changes version).

Reviewer #1 (Recommendations for the authors):1. I would recommend that the abbreviation SOAT is also included. ACAT can be confused with acetyl-coenzyme A acetyltransferase.

We have now included both SOAT and ACAT as abbreviations at the first instance of the mention of this enzyme in the Introduction (lines 70-71), as well as at several other points in the Results, Figures, and Materials and methods sections.

2. Page 5 & 6 and Figure 1, it would be much easier if the authors used simple abbreviations, 4βHC, 7αHC, 19HC rather than A, B, C, etc.

While it is unwieldy to list the abbreviated oxysterol names in the figure, we have included on page 6 the simple abbreviations for the 5 oxysterols that activate ACAT and diminish accessible PM cholesterol – 20(*R*)HC, 24(*R*)HC, 24(*S*)HC, 25HC, and 27HC (lines 165-166).

3. Figure 1. 20(R)-Hydroxycholesterol is an easier chemical nomenclature to follow than 20α-hydroxycholesterol. For completeness, I would define the stereochemistry at C25 in 27-hydroxycholesterol.

Thanks, we have now made these changes.

4. In the caption of Figure 4 it might be worth adding a note that a total of 25-HC was measured as a surrogate for free 25HC which I assume is the bioactive molecule. Or perhaps my assumption is wrong.

The 25HC measurements conducted in Figure 4A were for free 25HC (no esterase treatment). This detail has been included in the Figure Legend now (line 1526 in tracked changes version).

5. One thing I always wonder about is the oxysterol content of the medium, I guess it is low compared to the added 25HC, but I am guessing and don't really know.Overall, a really nice paper that was a pleasure to read!

The reviewer is correct in their guess that the oxysterol content of the medium is low compared to the added 25HC. Measurements of 25HC in control media samples (DMEM/F12 with 10% (v/v) FCS, 100 units/ml penicillin and 100 µg/ml streptomycin sulfate) gave values ranging from 1-20 nM, well below the treatment concentrations that were in the μM range.

Reviewer #2 (Recommendations for the authors):1. The paper only discusses cholesterol in the plasma membrane and the ER. However, other compartments, including the Golgi and the endocytic recycling compartment (ERC) also contain cholesterol. In CHO cells the ERC contains over 30% of the cellular cholesterol (DOI 10.1074/jbc.M1088612). There are also papers describing rapid (t_1/2_ 15 min) transport among these organelles, which is probably relevant for understanding the results in this paper. For example, cyclodextrin reduction of cholesterol will be compensated by cholesterol taken from several organelles – not just the ER. The rapid re-equilibration of cholesterol among membranes is not surprising given the rapid transport mediated by various sterol transporters.

We agree with the reviewer that intracellular cholesterol trafficking involves organelles other than the PM and the ER such as the Golgi and the endocytic recycling compartment. We have chosen to keep our focus on the PM and the ER because the effect of 25HC studied here is depletion of accessible cholesterol from the PM and two of the 25HC targets that mediate this depletion (ACAT and SREBPs) are located in the ER. We also agree that the rapid re-equilibration of cholesterol among membranes such as the PM and ER has been known for a long time, although we were struck by how potently this process depends on ACAT activity. We had referenced 4 out of the many previous studies that have contributed to our understanding of intracellular cholesterol trafficking (Refs. 16, 48-50). In this revised submission, we have added an additional reference to an informative review on this subject (PMID: 19286471; Ref. 51). We have also included a sentence in the Discussion addressing the possibility that 25HC treatment could also deplete accessible cholesterol from other organelles (lines 547-548 in tracked changes version).

2. Several of the figures show gels, and comparisons are made among conditions. It would be very helpful to have some quantification of these results and some indication of reproducibility.

We have now quantified all the Western blots in our paper and presented these results in 12 new data panels in 6 new Figure supplements (associated with main Figures 1, 2, 3, and 4). In all cases, the quantifications from 3-5 replicate experiments confirm the interpretations that we had made based on the representative Western blots.

3. For Figure 4, they used 5 μM 25HC, which is a high dose and probably is responsible for the long-lasting effects. At 8 hours, there is still about half the initial dose of 25HC in the cells, and they showed in Figure 1 that this would have a significant effect. It is likely that this amount of 25HC will still stimulate ACAT, preventing the cells from recovering. This hypothesis is easily tested with an ACAT inhibitor administered during the chase period. Thus, it is not just the continued effect on SREBP that is important.

We thank the reviewer for suggesting this elegant experiment, the results of which are shown in a new Figure 4C. We treated CHO-K1 cells with 5 µM of 25HC for 4 h, which completely abolished accessible cholesterol from the PMs (compare lane 2 to lane 1). We then removed the 25HC-containing media, washed the cells, and then added back media without 25HC. Similar to what we noted previously, no replenishment of accessible PM cholesterol was detected even after a 22 h recovery period (top two panels of Figure 4C). However, when we included 10 µM of an ACAT inhibitor (SZ58-035) during this recovery period, we began to detect accessible PM cholesterol after 8 h and ~60% of the initial level of accessible cholesterol was restored after 22 h (bottom two panels of Figure 4C and quantification in Figure 4 —figure supplement 2B). This result confirms the reviewer’s prediction that continued activation of ACAT by the lingering intracellular 25HC also plays a role in depleting accessible PM cholesterol over the longer time periods.

We have now carefully combed through our manuscript and modified all instances where we ascribed the long-term effects solely to 25HC’s effects on suppressing SREBP activation to also reflect ACAT’s contribution to the long-term depletion effects (*tracked changes* in revised manuscript). We are grateful to the reviewer for raising this point as it has helped clarify our mechanistic interpretations.

4. They mention the effects on macrophages at the bottom of page 13. This could be tested easily with ACAT inhibitors in macrophages.

In this paper, we have used CRISPR-Cas9 knockout models to show that 25HCmediated depletion of accessible PM cholesterol is dependent on ACAT activity in hamster cells (CHO-K1) and a human hepatocyte cell line (Huh7.5). To answer the reviewer’s question about whether a similar ACAT-dependence operated in macrophage cells, we used a commercially available ACAT inhibitor (SZ58-035). We first tested whether SZ58-035 would inhibit ACAT activity in RAW macrophage cells (mouse). Using our standard ACAT activity assay (described in Figure 1F), we found that 25HC stimulated ACAT activity was partially reduced from 4.2 nmol/mg/h to 1.4 nmol/mg/h when the cells were pre-treated with 10 µM SZ58-035 (see Author response image 1 – left 4 columns). Treatment with higher concentrations of SZ58-035 resulted in cell death. In contrast to the results with RAW cells, we observed almost complete inhibition of ACAT activity by 10 µM SZ58-035 in CHO-K1 cells, from 6.0 to 0.1 nmol/mg/h (Author response image 1 – right 4 columns).

Even though SZ58-035 could not completely inhibit ACAT activity, we tested whether its partial inhibition of ACAT would affect 25HC’s ability to deplete accessible cholesterol from RAW cell PMs. When the macrophage cells were treated with 25HC In the absence of SZ58-035, we observed a dose-dependent reduction in accessible cholesterol levels (similar to other cell types) (Author response image 1 – top 2 panels). In the presence of the ACAT inhibitor, we observed that 25HC’s effects were greatly reduced, suggesting that even partial inhibition of ACAT activity was sufficient to inhibit 25HC’s effects in macrophage cells (Author response image 1 – bottom 2 panels). While this result is suggestive, a rigorous answer to the reviewer’s question will require generation of a macrophage cell line where both ACAT1 and ACAT2 have been eliminated by CRISPRCas9 methods or even better would be a comparison of bone marrow-derived macrophages (BMDMs) from ACAT1^+/+^/ACAT2^+/+^ and ACAT1^-/-^/ACAT2^-/-^ mice. We are currently working on generating these models and will report the results in a future study.

5. In Figure 7, it is confusing to change the scale for each panel.

We agree, we have now changed the y-axis scales so they are the same.

Reviewer #3 (Recommendations for the authors):A point for discussion or future investigation could be that the lingering effect of 25-OH cholesterol is not completely defined as to whether this reflects an ongoing effect of 25-OH cholesterol to bind Insigs and inhibit cholesterol biosynthesis or an effect on ACAT.

This is an excellent point and we now present a new experiment in Figure 4C which shows that the lingering effect of 25HC involves ACAT (as well as Insigs). In this experiment, we treated CHO-K1 cells with 5 µM of 25HC for 4 h, which completely abolished accessible cholesterol from the PMs (compare lane 2 to lane 1). We then removed the 25HC-containing media, washed the cells, and then added back media without 25HC. Similar to what we noted previously, no replenishment of accessible PM cholesterol was detected even after a 22 h recovery period (top two panels of Figure 4C). However, when we included 10 µM of an ACAT inhibitor (SZ58-035) during this recovery period, we began to detect accessible PM cholesterol after 8 h and ~60% of the initial level of accessible cholesterol was restored after 22 h (bottom two panels of Figure 4C and quantification in Figure 4 —figure supplement 2B). This result confirms the reviewer’s prediction that continued activation of ACAT by the lingering intracellular 25HC also plays a role in depleting accessible PM cholesterol over the longer time periods.

We have now carefully combed through our manuscript and modified all instances where we ascribed the long-term effects solely to 25HC’s effects on suppressing SREBP activation to also reflect ACAT’s contribution to the long-term depletion effects (*tracked changes* in revised manuscript). We are grateful to the reviewer for raising this point as it has helped clarify our mechanistic interpretations.

In Figure 4D did ACAT1 KO prevent the effect of 25-OH cholesterol on the suppression of nuclear SREBP2? Would there be a way to more directly implicate sustained binding of low concentrations of 25-OH cholesterol to Insigs?

We did this control experiment and it is shown in Figure 2 —figure supplement 3. When ACAT1 KO cells were depleted of sterols, SREBP2 was processed to its nuclear form to a similar degree as wild-type cells (lane 1). When 25HC was added to these sterol-depleted cells, SREBP2 processing was suppressed in both cell lines with similar dose dependences (lanes 2 – 6). This indicates that 25HC enters the ACAT1 KO cells, reaches their ER membranes, binds to Insigs, and prevent SREBP cleavage in a similar fashion as in WT cells. In the experiment of Figure 4D, there is no initial depletion of accessible cholesterol by 25HC treatment for 4 h in ACAT1 KO cells and there is no further change after the 25HC is removed. The lingering 25HC’s suppression of SREBPs through binding to Insigs is not sufficient to reduce accessible cholesterol levels over the time period studied here.

It would also be helpful to know if concentrations of 25-OH cholesterol used in experiments or observed in cells are relevant to those likely to be achieved in the response to infectious agents in vivo.

We would also love to know the amount of 25HC generated by a cell during infection, but as the reviewers can appreciate, measuring the concentration of 25HC in a particular cell is technically challenging. Even so, we can make some estimates based on measurements reported in the literature by us and others.

Two different studies (PMID: 32284563 and PMID: 23273843) reported that upon interferon-g stimulation (25-500 U/ml for 12-24 h), the amount of 25HC secreted by bone marrow-derived macrophages (BMDMs) into the extracellular media accumulated to a final concentration of ~0.4 µM. It is plausible that the concentration of 25HC inside of the 25HC-producing cell is much higher than the 0.4 µM value measured upon dilution into the extracellular medium.

One of the previous studies (PMID: 23273843) did also measure the intracellular 25HC produced after interferon-g stimulation and reported a value of 0.2 nmol per 10^6^ cells. Another group conducted a morphological analysis of macrophages and reported that their volume was 1000-5000 µm^3^ (PMID: 9400735). Using these estimates, the intracellular concentration of 25HC upon interferon-g stimulation could range from 40-200 µM.

The concentration of 5 µM that we use in many of our studies is between these bookend values and thus may not be completely unreasonable. We look forward to future technical breakthroughs which will allow a more precise answer to this question.

Although the authors provide a focused and convincing discussion on the role of PM cholesterol in infectious processes, there are broader aspects of 25-OH cholesterol biology that are relevant to infectious processes that could be discussed and the topic broadened. In particular, Reboldi et al. (Science 2014) showed that 25-OH cholesterol acts to suppress the priming and activity of inflammasomes. This results in reduced inflammatory death but increased susceptibility to infection with L. monocytogenes. Although not necessarily contradictory, it would be interesting for the authors to discuss these earlier results in the context of their own findings.

We thank the reviewer for raising this point. We have referenced a review by Jason Cyster that covers this work in detail (Ref. 5). We agree the topic of how 25HC could exhibit distinct immunomodulatory activities is of great interest, and likely reflects the diverse cell types infected by pathogens and the immune regulatory components expressed such as pattern recognition receptors, signaling components, and inflammasome subtypes. Importantly, we are currently planning to test if ACAT stimulation and changes in PM accessible cholesterol could contribute to inflammasome activation in response to various pathogenic insults.

Specific PointsLine 96. Please cite the original report (Costet P et al., JBC 2000).

Thanks for pointing out our oversight, we have now included this reference (Ref. 26).

Lines 227-231. In Figure 2A ALOD4 binding seems somewhat reduced by 25-OH cholesterol in ACAT KO cells, especially after correction for actin. The same seems true when inactive ACAT is transected in the cells (bottom rows). How do the authors explain this? In Figure 2B the fOlyA fluorescence seems quite variable.

Please see Figure 2 —figure supplement 2 where we present the quantification of the Western blots in Figure 2A and two other replicate experiments. The quantifications confirm the visual interpretations that we had made based on the representative Western blots. We agree that the fOlyA fluorescence is in general more variable than that for ALOD4, we have not investigated the causes for this.